

# Quantifying the climatic impact of crude oil pollution on sea ice albedo

**Benjamin Heikki Redmond Roche[1] and Martin Daniel King[1]**

[1]Department of Earth Sciences, Royal Holloway, University of London, Egham, Surrey, TW20 0EX, UK

*Correspondence to*: Professor Martin King (m.king@rhul.ac.uk)

**Abstract.** Sea ice albedo plays an important role in modulating the climate of the Earth and is affected by low background concentrations of oil droplets within the ice matrix that absorb solar radiation. In this study the albedo response of three different types of sea ice (melting, first-year, and multi-year sea ice) are calculated at increasing mass ratios (0–1000 ng g$^{-1}$) of crude oil by using a coupled atmosphere-sea ice radiative-transfer model (TUV-snow) over the optical wavelengths 400–700 nm. The different types of quasi-infinite thickness sea ice exhibit different albedo responses to oil pollution, with a 1000 ng g$^{-1}$ mass ratio of oil causing a decrease to 70.9% in multi-year sea ice, 47.9% in first-year sea ice, and 22% in melting sea ice relative to the unpolluted albedo at a wavelength of 400 nm. The thickness of the sea ice is also an important factor, with realistic thickness sea ices exhibiting similar results, albeit with a weaker albedo response for multi-year sea ice to 75.3%, first-year sea ice to 66.7%, and melting sea ice to 35.7%. The type of oil also significantly affects the response of sea ice albedo, with a relatively opaque and heavy crude oil (*Romashkino* oil) causing a significantly larger decrease in sea ice albedo than a relatively transparent light crude oil (*Petrobaltic* oil). The size of the oil droplets polluting the oil also plays a minor role in the albedo response, with weathered submicron droplets (0.05–0.5 µm radius) of *Romashkino* oil being the most absorbing across the optical wavelengths considered. Therefore, the work presented here demonstrates that low background concentrations of small submicron to micron-sized oil droplets have a significant effect on sea ice albedo. All three types of sea ice are affected, however first-year sea ice and particularly melting sea ice are very sensitive to oil pollution; thus, the Arctic may become more vulnerable to oil pollution as the ice becomes progressively thinner and younger in response to a changing climate.

## 1 Introduction

Arctic sea ice has significantly declined from its 1981–2010 spring and summer averages, both in extent and thickness of sea ice cover (Fetterer et al., 2017). The summer sea ice minimum has decreased 13.1% per decade from the 1981–2010 average, with an average extent of 6.85 million km$^2$ in 1979–1992 compared to an average extent of 4.44 million km$^2$ from 2007–2020 (Thoman et al., 2020). Perennial sea ice cover decline is between 12.2% and 13.5% for first-year sea ice, and 15.6% and 17.5% for multiyear sea ice per decade, respectively (Comiso, 2012; Tschudi et al., 2019). Consequently, it is now very likely that an ice-free Arctic Ocean, a so called 'Blue Ocean Event', will be realised by the mid-century unless there is a rapid reduction in greenhouse gas emissions (Notz and Stroeve, 2018). In response to the 'blue' Arctic Ocean, there has been a significant interest in developing northern shipping routes which can decrease journey lengths from Europe to Asia by up to 40% (Ho, 2010; Eguíluz et al., 2016; Kikkas and



Romashkina, 2018). Additionally, 13% of total global undiscovered oil reserves are estimated to be in the Arctic Ocean and their exploitation is of great geopolitical importance for the Arctic (Bird et al., 2008; Krivorotov and Finger, 2019; Czarny, 2019). Therefore, well established concerns prevail for the associated impacts that blowouts, pollution from offshore drilling, production, and transportation of oil
may have in the Arctic (Koivurova and Vanderzwaag, 2007; WWF-Canada, 2011; Gulas et al., 2017).

Sea ice albedo plays a key role in modulating the climate of the Earth. The high latitude, radiative balance, is primarily controlled by shortwave solar radiation which significantly effects both sea ice and snow cover in the region (e.g. Perovich et al., 1998; Flanner et al., 2007). Several different physical properties affect sea ice albedo, the most significant of which are: density, grain size, grain shape, brine
content, thickness, snow cover, and light-absorbing impurities (e.g. Perovich et al., 1998; Perovich, 2003; Marks and King, 2014; Hancke et al., 2018). The wavelength integrated and spectral albedos for different types of sea ice have previously been considered (e.g. Grenfell and Makyut, 1977; Grenfell and Perovich, 1984; Pervoich et al., 1986; Buckley and Trodahl, 1987; Grenfell, 1991; Perovich, 1996; Hanesiak et al., 2001); this study focuses on three types of sea ice: melting, first-year, and multi-year sea ice. The optical
properties of these different types of sea ice are described in Marks and King (2014) and Lamare et al (2016). The albedo of sea ice is wavelength dependent with maximum albedo values occurring at 390 nm in pure ice, where absorption is at a minimum (Warren at al., 2006). The absorption of light absorbing impurities are also wavelength dependent and affect where the maximum albedo occurs depending on their absorption spectra and the amount of the impurities that are contained within the ice. Different light
absorbing impurities in sea ice such as volcanic dust, mineral dust, black carbon and soot have previously been examined for their effect on albedo (e.g. Warren and Wiscombe, 1980; Warren, 1984; Light et al., 1998; Doherty et al., 2010; Marks and King, 2013, 2014; Lamare et al., 2016; Marks et al., 2017) and were found to have significant effects on sea ice albedo, even at very low concentrations. Indeed, Glaeser and Vance (1971) released oil on top of ice and found that oil absorbs 30% more heat from the sun than
normal ice. However, aside from limited and extreme field studies by NORCOR (1975), and Gavrilo and Tarashkevich (1992), the effects that oil pollution has upon sea ice albedo have not previously been considered in literature, so this is explored here in a modelling study.

As oil is released into sea water it is influenced by several weathering processes: evaporation, dispersion, wave action, sedimentation, photo-oxidation and bioremineralisation (e.g. Daling et al., 1990;
Resby and Wang, 2004; Dillipiane et al., 2021). The physicochemical properties of oil (e.g., water-in-oil emulsion viscosity, density, pour point) are constrained by oil composition, wave energy, and ice conditions, which largely determine the fate of oil in cold marine environments (Brandvik and Faksness, 2009; Brandvik et al., 2010; Singsaas et al., 2020). In low energy environments where nonbreaking waves occur, oil spilled at the surface will spread into a slick owing to gravity, viscosity, and surface tension. In
high energy environments where plunging, spilling and breaking waves occur, oil droplets are entrained into the water and continually resurface, resulting in the droplet size decreasing (e.g. Delvigne and Sweeney, 1988; Wang et al., 2005; Z. Li et al., 2008a, 2008b; C. Li et al., 2017; Wilkinson et al., 2017). In these high energy conditions, oil can spread several square kilometres in several hours and several hundred square kilometres within several days (Berenshtein et al., 2020). There have been extensive
modelling studies into both large-scale submarine blowouts (e.g. Johansen, 2003; Zheng et al., 2003; Lima Neto et al., 2008; Socolofsky et al., 2008; Fraga et al., 2016; Dissanayake et al., 2018) as well as surface oil slicks (e.g. Spaulding et al., 1992; Reed and Rye, 1995; Daling et al., 1997; Papadimitrakis et



al., 2005, 2011; Gamzaev, 2009), with the consensus being that knowledge of oil droplet size distribution is fundamental to accurately model ocean oil spills (Nissanka and Yapa, 2018). There have also been several lab (Hesketh et al., 1991; Masutani and Adams, 2000; Tang and Masutani, 2003; Brandvik et al., 2013, 2014; Wang et al., 2018) and field experiments (Johansen et al., 2003; Brandvik et al., 2010) which have generally focused on very high levels of pollution within a relatively short time frame from the release of oil. However, away from the initial spill site the size of oil droplets dispersed by breaking waves is typically a log-normal distribution of smaller spherical particles (Otremba, 2007; Z. Li et al., 2011; Johansen et al., 2013; Haule et al., 2015; Haule and Freda, 2016).

Three major Joint Industry Programs into the behaviour and fate of oil in sea ice have taken place – the first was a four year SINTEF led project looking at 'Oil Spill Contingency for Arctic and Ice-covered Waters' (Sørstrøm et al., 2010). The second was a three year International Association of Oil and Gas Producers led project looking at 'Arctic Oil Spill Response Technology' (Dickins, 2017). The third was a three year collaboration between SINTEF and the Research Council of Norway looking at the 'Fate, Behaviour and Response to Oil Drifting into Scattered Ice and Ice Edge in the Marginal Ice Zone' (Singsaas et al., 2020). Smaller studies have looked into the hyperspectral features of oil-polluted sea ice (Liu et al., 2018) and the effects of exposure to crude oil on ice algae (Dilliplaine et al., 2021). In another study, Gavrilo and Tarashkevich (1992) found that crude oil can migrate vertically through multiyear ice if it is pumped either into or below the ice. They also found that the presence of oil decreased both the albedo and mechanical strength of the multiyear sea ice but did not measure the direct relative change in ice albedo in order to parameterise the effects. A similar study by NORCOR (1975) also concluded that the presence of oil released onto the ice may accelerate melting by 1 to 3 weeks but did not parameterise the effects either. Whilst it is now recognised that oil can be trapped in ice in a variety of ways (e.g. Dickins and Buist, 1981; Buist and Dickins, 1983; Buist et al., 1983; Drozdowski et al., 2011) what is missing from the literature is an understanding of how oil exists at low concentrations over a large spatiotemporal range in cold marine environments and what their climatic significance may be.

Studies from the Gulf of Thailand and South China Sea since the 1970's indicate that mass ratios of oil are highly variable and dependent on location and season but can be up to 75 ng g$^{-1}$ (75 ppb), even far offshore (Law and Mahmood, 1986; Wongnapapan et al., 1999). In regions of intensive shipping and marine transportation, particularly near to offshore oil fields, concentrations of oil ranging from several ppb to ppm are common (Haule and Freda, 2016), and oil concentrations of ship effluent discharge are only limited to a value of 15 ppm (i.e. 15,000 ng g$^{-1)}$) (IMO, MARPOL Annex I). Mega oil spills are also capable of transporting significant quantities of oil vast distances in sea water. Following the Deepwater Horizon blowout oil mass ratios exceeding 100 ng g$^{-1}$ were transported via currents over 1000 km from the spill site more than 60 days after the event occurred (Berenshtein et al., 2020). Therefore, it is important to consider a wide range of oil concentrations to replicate a variety of scenarios.

Oil droplets frequently weather more slowly and are more durable in cold environments than warm environments (Venkatesh et al., 1990; Singsaas et al., 2020). Smaller droplets (<70 μm diameter) are affected by physical processes induced by atmospheric forcing (e.g. shear-induced turbulence, sea-surface waves, Langmuir circulations, Ekman transport, and thermal convection) in the euphotic zone, resulting in the continuous mixing of the top of the water column and the formation of the ocean mixed layer (Edson et al., 2007; Sullivan and McWilliams, 2010; Özgökmen et al., 2012; D'Asaro, 2014; Xiao and Yang, 2020). The droplets remain in suspension for extended periods of time (i.e. weeks to months) in



the presence of adequate mixing energy, with their buoyancy acting as a constant resistant force to the mixing (Ryerson et al., 2012; Yang et al., 2014; Zhao et al., 2016; Uttieri et al., 2019; Xiao and Yang, 2020). The following papers (e.g. Lunel, 1993; Reed et al., 2009a, 2009b; Bandara and Yapa, 2011; Brandvik et al., 2013, 2021; Johansen et al., 2013; Zhao et al., 2014a, 2014b, 2016, 2017a, 2017b; North et al., 2015; Zeinstra-Helfrich et al., 2015a, 2015b, 2016; Nissanka and Yapa, 2016, 2017, 2018; Z. Li et

al., 2017a, 2017b; Wang et al., 2018) demonstrate the typical sizes of oil droplets in sea water (3–500 µm, locally up to 12,000 µm) and also focus on relatively large concentrations of crude oil representing the immediate area around a spill site, whereas the study presented here focuses on relatively low background concentrations of weathered crude oil over a large area. The focus of these studies is on droplet sizes between microns to millimetres which demonstrate a logarithmic dependence between number density

and droplet size that increases as droplet size decreases (Lunel, 1993; Brandvik et al., 2013, 2021; Johansen et al., 2013; Zhao et al., 2017b; Nissanka and Yapa, 2017; C. Li et al., 2017). These larger micron to millimetre sized droplets are characteristic of oil released in the subsurface, such as during a blowout or similar specific acute event of small geographical extent, which have not undergone significant weathering or transport (e.g. Johansen et al., 2013; Brandvik et al., 2013). The number density of smaller

sized droplets is potentially not reported owing to limitations of measuring equipment or techniques, and as pointed out by Otremba (2007), they are measuring the larger radius tail of a log-normal distribution of a number density against droplet radius. It is likely that the most numerous droplets affecting wide areas of polar sea ice are these smaller sized weathered oil droplets. The smallest droplets may not be considered in these studies owing to four factors: (i) the proximity and timing of measurements to droplet

release; (ii) equipment limitation in droplet size-range detection (e.g. LISST-100X below 2.5 µm) coupled with the effect of background noise at the smallest scale (e.g. Brandvik et al., 2013); (iii) droplets smaller than 0.2 µm being considered as dissolved (e.g. North et al., 2015); (iv) the difficulty and high uncertainty of direct measurements in situ (e.g. Gonçalves et al., 2016). In addition, Nissanka and Yapa (2017) suggest that many of the equilibrium models utilised in the literature (e.g. Delvigne and Sweeney, 1988;

Spaulding et al., 1992; Delvigne and Hulsen, 1994; Reed and Rye, 1995; Tkalich and Chan, 2002; Reed et al., 2009a, 2009b; Johansen et al., 2013, 2015) omit to consider the temporal modification of oil droplet size distributions owing to droplet break up and coalescence.

        The study presented here utilises a coupled atmospheric and sea ice radiative-transfer model to quantify how oil pollution may affect the albedo of three different types of sea ice (melting sea ice, first-

year sea ice, and multi-year sea ice). The principle aim of this study is evaluating the impacts increasing amounts of two crude oils that band an envelope of crude oils optically have on sea ice albedo. Previous studies have demonstrated that the size and absorption cross section of light absorbing impurities and the type of sea ice affect how sensitive a sea ice is to increasing mass densities of pollutant (e.g. Marks and King, 2013, 2014; Lamare et al., 2016; Marks et al., 2017). Therefore, the effect that oil has on the three

different types of sea ice and at different oil droplet sizes are also evaluated.

## 2 Methods

        The methodology is separated into three sections: a description of the radiative-transfer model, a description of the optical properties of the pollutants used in the model, and the different model runs that were examined in this study.



## 2.1 Radiative-transfer model

The Tropospheric Ultraviolet and Visible Radiation model was originally written by Madronich and Flocke (1998) and later adapted into a coupled atmosphere-snow/sea ice model (TUV-snow) by Lee-Taylor and Madronich (2002). The model is capable of calculating radiative-transfer parameters from the top of the atmosphere into the sea ice and snow layers found at the ocean-atmosphere interface, and has been used in several studies (e.g. King et al., 2005; Marks and King, 2013, 2014; Lamare et al., 2016; Marks et al., 2017). The model utilises an eight-stream discrete-ordinates 'DISORT' algorithm (Stamnes et al., 1988) with a pseudo-spherical correction, and is comprehensively described by Lee-Taylor and Madronich (2002). In this study the albedo response of three different types of sea ice (multi-year sea ice, first-year sea ice and, melting sea ice) to increasing amounts of two different crude oils are calculated using the TUV-snow model. The sea ice layers are assumed to be horizontally homogeneous, weakly absorbing and highly scattering. The optical properties of the sea ice layers are described by an asymmetry factor, $g$, a wavelength independent scattering cross-section, $\sigma_{scatt}$, a wavelength dependent absorption cross-section, $\sigma_{abs}$, and the sea ice density (Lee-Taylor and Madronich, 2002; France et al., 2011; Reay et al., 2012; Marks and king, 2013, 2014; Lamare et al., 2016; Marks et al., 2017). The optical and physical parameters of the three sea ices emanate from several field studies (Grenfell and Maykut, 1977; Perovich, 1990, 1996; Timco and Frederking, 1996; Gerland et al., 1999; Simpson et al., 2002; Fisher et al., 2005; King et al., 2005; France et al., 2011; Marks and King, 2013) and are summarised in Table 1.

**Table 1**. TUV-snow model sea ice input parameters derived from literature and detailed in Marks and King (2014).

| Type of sea ice | Density (kg m$^{-3}$) | Scattering cross section (m$^2$ kg$^{-1}$) | Asymmetry parameter ($g$) | Quasi-infinite ice thickness (cm) | Realistic ice thickness (cm) |
|---|---|---|---|---|---|
| Melting | 800 | 0.03 | 0.98 | 1500 | 250 |
| First-year | 800 | 0.15 | 0.98 | 1500 | 80 |
| Multi-year | 800 | 0.75 | 0.98 | 1500 | 250 |

The atmosphere and sea ice in the model are separated into 201 layers comprising a total thickness of 90 km and are shown in Table 2. The optical effects are considered both in vacuo, by using a quasi-infinite thickness sea ice, and at realistic thicknesses of sea ice. The sea ice is set to be optically thick at a quasi-infinite thickness of 15 m (Lamare et al., 2016) therefore making albedo changes at the ice-atmosphere interface independent of the layer under the sea ice and allowing for a fair comparison of the effects explored here. The realistic thickness sea ices are also considered to allow for a more realistic effect on the different sea ice albedos to be observed. The realistic multi-year and melting sea ice are assigned a thickness of 2.5 m and the first-year sea ice is assigned a thickness of 0.8 m (Bourke and Garrett, 1987; Laxon et al., 2003; Weeks, 2010).

**Table 2**. TUV-snow model sea ice input parameters derived from literature and detailed in Marks and King (2014).





| Ice thickness (m) | Number of layers | Layer structure |
|---|---|---|
| 0.8 | 141 | 1 cm increments 0–0.1 m; 10 cm increments 0.1–0.2 m; 2 cm increments 0.2–0.6 m; 1 cm increments 0.6–0.7 m; 1 mm increments 0.7–0.8 m |
| 2.5 | 141 | 1 cm increments 0–0.1 m; 10 cm increments 0.1–2.2 m; 2 cm increments 2.2–2.4 m; 1 mm increments 2.4–2.5 m |
| 15 | 141 | 1 cm increments 0–0.1 m; 10 cm increments 0.1–1 m 1 m increments 1–14 m; 10 cm increments 14–14.9 m; 1 mm increments 14.9–15 m |
| Atmosphere (90 km thick) | 60 | (10 cm increments 0.8–1 m; 4.5 m increments 1–10 m; 10 m increments 10–100) (1.875 m increments 2.5–10 m; 10 m increments 10–100 m;) 1 m increments 15–20 m; 10 m increments 20–100 m; 100 m increments 100–1000 m; 1 km increments 1–10 km; 2 km increments 10–42 km; 4 km increments 42–90 km |

The 'flat plate' irradiance reflectance of the surface layer of the sea ice is calculated in the model and is referred to as albedo in the work described here, represented by Eq. (1):

$$A = \frac{E_u}{E_d},$$ (1)

Where $E_u$ is the upwelling irradiance and $E_d$ is the downwelling irradiance. The oil droplets in this study are considered to have a log-normal size distribution that is representative of weathered oil according to experimental measurements by Otremba (2007). The log-normal distribution is centred on a
droplet radius size of 0.05 µm and spans a range of 0.05–5 µm, in line with Haule and Freda (2016). Furthermore, a wide range of oil mass ratios are considered (0–1000 ng g$^{-1}$) so that the data is applicable for a variety of different scenarios. The wavelength independent albedo of the layer under the sea ice is fixed to 0.1, representative of a dark ocean layer (Payne, 1972). The sea ice is illuminated isotropically to allow ready comparison with previous work (e.g. Marks and King, 2013, 14; Lamare et al., 2016;
Marks et al., 2017) and remove a solar zenith angle dependency. As shown elsewhere (e.g. Marks and King, 2013), the isotropic illumination allows any solar zenith angle dependence to be disregarded for simplicity. The atmosphere is assumed to be aerosol free with a typical ozone column of 300 Dobson units. The distance between the Earth and the Sun is set to 1 astronomical unit. The model produced a wavelength dependent surface albedo at a 1 nm interval over a spectrum covering 400–700 nm,
corresponding to the wavelength range of the available oil optical properties and typical optical range measured in the field.



## 2.2 Optical properties of crude oil

The model utilises the absorption spectra for ice as defined in Warren and Brandt (2008). The under-ice surface, snow cover, average Arctic black carbon (5.5–19.2 ng g$^{-1}$), and average aerosol mass
ratios (≤193.2 ng g$^{-1}$) are not considered for the quasi-infinite sea ice in order to maintain an independent albedo change (Zdanowicz et al., 1998; Grenfell et al., 2002 Jiao et al., 2014). However, a very low background mass ratio of black carbon (1ng g$^{-1}$) is included, in line with previous studies (e.g. Marks and King, 2013; 2014), and representative of the most pristine snowpack away from the stations at both Dome C, Antarctica and Summit, Greenland (Zatko et al., 2013). For the realistic thickness sea ices, a typically
low Arctic background black carbon mass ratio of 5.5 ng g$^{-1}$ is selected (Grenfell et al., 2002). The droplets are evenly distributed throughout the ice to be representative of seasonal oil movement in the ice (NORCOR, 1975; Martin, 1979; Drozdowski et al., 2011). The mass ratio of oil in the ice is small and therefore the light scattering properties of the ice matrix dominate, thus only the absorption of light by the oil droplets is considered. The total absorption in sea ice is represented by Eq. (2):

$$\sigma_{\text{total}} = \sigma_{ice}(\lambda) + \chi_{oil}\sigma_{oil}(\lambda) \tag{2}$$

Where $\sigma_{ice}(\lambda)$ is the wavelength-dependent absorption cross section of sea ice, $\chi_{oil}$ is the mass ratio of crude oil in the sea ice, and $\sigma_{oil}$ is the wavelength dependent mas absorption cross-section by
crude oil.

The optical properties of several types of crude oil, lubricating oil and petroleum are explored by Otremba (2000), who parameterised the real and imaginary parts of their refractive index. Otremba established that, for the oils capable of forming a surface film, *Romashkino* crude oil has the largest imaginary refractive index and absorption coefficient values whereas *Petrobaltic* crude oil has the
smallest values, with other oils bracketed by these parameters (Otremba, 2000). *Petrobaltic* is described as a relatively transparent light crude oil, whereas *Romashkino* is described as a heavy, relatively opaque crude oil (Otremba, 2005; 2007; De Carolis et al., 2014). Whilst both crude oils have a variety of uses, including as marine engine fuels, *Romashkino* can be considered a typical marine engine Heavy Fuel Oil (De Carolis et al., 2014; IMO, MARPOL Annex 1). In the model *Petrobaltic* and *Romashkino* are
assigned characteristic light and heavy crude oil densities of 0.77 g cm$^{-3}$ and 0.94 g cm$^{-3}$, respectively (Hollebone, 2015). Both mechanically and chemically dispersed small droplets tend to be spherical in nature, so they are considered to be spherical in the model (Johansen et al., 2013). A log-normal distribution of number density versus radius with a peak radius of 0.05 μm is utilised, corresponding to the weathered oil experimentally measured in Otremba (2007) and used in other studies (e.g. Haule et al.,
2015; Haule and Freda, 2016). The log-normal distribution is represented by Eq. (3):

$$\frac{dN}{d\ln(r)} = \frac{No}{\sqrt{2\pi}} \frac{1}{\ln(s)} \exp\left(\frac{-(\ln(r)-\ln(r_m))^2}{2\ln^2(s)}\right) \tag{3}$$

Where $r$ is the droplet radius, $r_m$ is the median radius of the size distribution, $s$ is the geometric
standard deviation, $N_o$ is the total particle density, and $N$ is the particle density (Seinfeld and Pandis, 2006). To quantify the size effect a range of droplet sizes spanning 0.05–5 μm (Haule and Freda, 2016)



are considered, and the geometric standard deviation of the mean is taken from Otremba (2007). Owing to the size of the droplets and the range of the spectrum considered (400–700 nm), Mie calculations (Bohren and Huffman, 1983) are required to record the efficiency of light absorption for particles of
different sizes within the Mie regime, in a sea ice or brine matrix. There are two similar modern determinations of the refractive index of ice (Warren and Brandt, 2008; Picard et al., 2016), to stay consistent with previous work (e.g. Marks and King, 2013; 2014) the values of Warren and Brandt (2008) are used in this paper. The real refractive index of the brine was 1.3402 at a wavelength of 600 nm (Quan and Fry, 1995) and set to a temperature of 0º C and a salinity of 35‰, the smallest temperature and largest
salinity values available.

Mie calculations of spherical oil droplets are weighted by log-normal size distributions using a modified version of the 'BHMIE' computational code (Bohren and Huffman, 1983). The calculations utilise the real (0º C) and imaginary refractive index values of *Romashkino* and *Petrobaltic* oil from Otremba (2000) and are applied in the model in agreement with literature (Otremba, 2007; Haule and
Freda, 2016). The Mie calculations compute the mass absorption cross sections of the oil from the absorption efficiencies. The mass absorption coefficients of the oils are used in the TUV-snow computational code (Lee-Taylor and Madronich, 2002) to calculate the effect that increasing mass ratios of *Romashkino* and *Petrobaltic* oil have on the albedo of the different types of sea ice.

**2.3 Calculation of albedo on sea ice with different loadings of crude oil**

Albedos of quasi-infinite first-year sea ice doped with different mass ratios of both *Petrobaltic* and *Romashkino* from 0–1000 ng g⁻¹ (0, 5, 10, 25, 100, 150, 200, 250, 300, 350, 400, 450, 500, and 1000 ng g⁻¹) are calculated as a function of wavelength. The albedos of quasi-infinite melting, first-year, and multi-year sea ice are then calculated at increasing mass ratios of both *Petrobaltic* and *Romashkino* from 0–1000 ng g⁻¹. The effect of oil droplet size (in the range 0.05–5.0 μm) are then considered for a single
wavelength (400 nm) for the oil with largest absorption cross section and for quasi-infinite sea ice with the largest penetration depth to light (Marks and King, 2014). These conditions produce the largest variation in sea ice albedo with changing droplet size. The mass absorption coefficient of droplets (in the range 0.005–50 μm) are also considered for the wavelengths 400–700 nm. The albedo of quasi-infinite melting sea ice polluted with increasing amounts of black carbon (1–100 ng g⁻¹) and *Romashkino* oil are
also considered to investigate how sea ice albedo responds to increasing mass ratios of oil as a function of background pollution of sea ice. The albedos of realistic thickness melting, first-year, and multi-year sea ice are also calculated at increasing mass ratios of both *Petrobaltic* and *Romashkino* from 0 to 1000ng g⁻¹.

**3 Results**

The results are separated into five sections: how the type of oil and increasing mass ratios of oil effect quasi-infinite sea ice albedo, the sensitivity of each type of quasi-infinite sea ice to increasing oil, the effect of droplet size distribution on quasi-infinite sea ice albedo, the impact that increasing background mass ratios of black carbon have on the further reduction of quasi-infinite sea ice albedo by oil, and the effect oil has on each type of realistic thickness sea ice.




## 3.1 The effect of increasing mass ratios of different types of oil on the albedo of quasi-infinite sea ice


Figure 1 illustrates the surface albedo of quasi-infinite first-year sea ice between 400–700 nm as a function of *Petrobaltic* and *Romashkino* oil mass ratios that are evenly distributed throughout the ice and increase from 0–1000 ng g$^{-1}$. As the mass ratios of both oils increases sea ice albedo decreases. The

oil is most absorbing at 400 nm, where ice is the least absorbing (Warren and Brandt, 2008). The albedo at 400 nm with no oil pollutant is 0.87, with an addition of 10 ng g$^{-1}$ of *Romashkino* resulting in a decrease in albedo to 97.4% of the unpolluted albedo and 10 ng g$^{-1}$ of *Petrobaltic* to 99.8%. Therefore, the *Romashkino* is more absorbing per unit mass than the *Petrobaltic*, in line with Otremba (2000), and the albedo decreases dramatically with increasing mass ratios of oil. For instance, at a mass ratio of 100 ng

g$^{-1}$ *Romashkino* and *Petrobaltic* cause an albedo decrease to 83.8% and 98.3% of the unpolluted albedo, respectively, and at 1000 ng g$^{-1}$, *Romashkino* and *Petrobaltic* cause a decrease to 47.1% and 88.2% of the unpolluted albedo, respectively. The albedo response to increasing mass ratios of oil is non-linear (Fig. 2) and is exemplified by the doubling of *Romashkino* mass ratios from 100 to 200 ng g$^{-1}$ causing a decrease in albedo to 89.8% of the 100 ng g$^{-1}$ value, whilst doubling of the mass ratios from 500 to 1000 ng g$^{-1}$

causes a decrease in albedo to 77.9% of the 500 ng g$^{-1}$ value. For *Petrobaltic* there is also a non-linear response as mass ratios increase but it is significantly less pronounced (98.5% and 94.8% of the unpolluted albedo, respectively).

The change in albedo is wavelength dependent, and the effect of oil significantly decreases as wavelength increases over the region studied and the oil becomes less absorbing whilst the ice becomes

more absorbing. At 550 nm 1000 ng g$^{-1}$ of *Romashkino* causes albedo to decrease to 81.6% of the unpolluted albedo, and at 700 nm this decrease is to 95.5% of the unpolluted albedo. The effect *Petrobaltic* has on albedo above 500 nm is negligible, even at larger mass ratio. Therefore, at shorter wavelengths (400–550 nm) the albedo is more responsive to the oil whereas at longer wavelengths (550–700 nm), the albedo is more responsive to the ice.




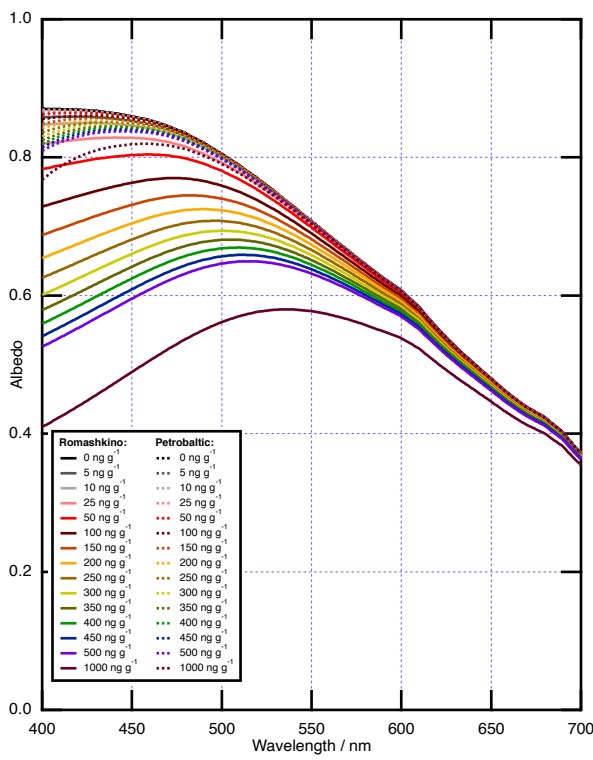

**Fig. 1.** Wavelength dependent albedo of first-year sea ice at increasing mass ratios of *Romashkino* (solid) and *Petrobaltic* (dotted) oils (0–1000 ng g$^{-1}$) with a median droplet radius of 0.05 μm. The first-year sea ice is quasi-infinite (15 m thick), and the properties are described in Table 1. These data demonstrate sea ice albedo is significantly more responsive to *Romashkino* oil than *Petrobaltic* oil.

### 3.2 The sensitivity of each type of quasi-infinite sea ice to increasing oil

Figure 2 illustrates the surface albedo response of quasi-infinite multi-year, first-year, and melting sea ice as mass ratios of *Romashkino* and *Petrobaltic* oils increase from 0–1000 ng g$^{-1}$. The three types of sea ice have different unpolluted albedos: melting sea ice 0.72, first-year sea ice 0.87, and multi-year sea ice 0.94 at a wavelength of 400 nm respectively, owing to their different scattering cross sections (Perovich, 1996; Marks and King, 2014). The three types of sea ice also respond differently to increasing mass ratios of oil, analogous to previous studies exploring the response of sea ice to mineral dust (Lamare et al., 2016). As the mass ratio of *Romashkino* oil increases in multi-year sea ice the albedo decreases to 98.8% at 10 ng g$^{-1}$, 92.3% at 100 ng g$^{-1}$, and 70.9% at 1000 ng g$^{-1}$ of the unpolluted albedo at a wavelength of 400 nm. In first-year sea ice, the albedo decreases to 97.4% at 10 ng g$^{-1}$, 83.8% at 100 ng g$^{-1}$, and 47.1% at 1000 ng g$^{-1}$ of the unpolluted albedo at a wavelength of 400 nm. In melting sea ice, the albedo decreases to 94.9% at 10 ng g$^{-1}$, 68.8% at 100 ng g$^{-1}$, and 22% at 1000 ng g$^{-1}$ of the unpolluted albedo at a wavelength of 400 nm. The same response occurs with *Petrobaltic* oil pollution, although the effect is much weaker, with multi-year sea ice albedo decreasing to 94.5% at 1000ng g$^{-1}$; first-year sea ice albedo decreasing to 88.2% at 1000 ng g$^{-1}$; and melting sea ice albedo decreasing to 77% at 1000 ng g$^{-1}$ of the





unpolluted albedo at a wavelength of 400 nm. Therefore, the most responsive type of sea ice to oil pollution is melting sea ice whilst multi-year sea ice is the least responsive.

At longer wavelengths, the three types of sea ice are less responsive to increasing mass ratios of either *Romashkino* or *Petrobaltic* as the absorption coefficient of the ice increases (Warren, 2019). At a
345 wavelength of 500 nm increasing mass ratios of *Romashkino* oil significantly effect albedo, however at wavelengths longer than 500 nm the effect of oil pollution is negligible for the three types of sea ice. Evidently the albedos of the three types of sea ice are very wavelength dependent and the response in albedo to increasing mass ratios of oil significantly decreases as ice absorption increases at longer wavelengths (Warren and Brandt, 2008; Warren, 2019).
350 Both the type of oil and the type of ice are significant in changing the albedo. There is a large difference in how responsive each type of sea ice is to increasing mass ratios of oil and melting sea ice is significantly more affected by oil pollution than multi-year sea ice. There is also a significant difference in the effect that different oils have on sea ice albedo. At a wavelength of 400 nm and a relatively low increase in mass ratios of oil (10–100 ng g$^{-1}$), *Romashkino* oil causes a relative decrease in albedo to
355 72.5% in melting sea ice, 86% in first-year sea ice, and 93.4% in multi-year sea ice. At the same wavelength and increasing mass ratios of *Petrobaltic* oil, there is a relative decrease in albedo to 99.3% in multi-year sea ice, 98.4% in first-year sea ice, and 97% in melting sea ice. Therefore, as mass ratios of oil increase, it is the type of oil that has a larger effect on how responsive sea ice albedo is, rather than what the type of ice is.





**Fig. 2.** Albedos of multi-year, first-year, and melting sea ice with increasing mass ratios of *Romashkino* (A, B, and C) and *Petrobaltic* (D, E, and F) oils (0–1000 ng g⁻¹) with a median droplet radius of 0.05 μm. The selected wavelengths are 400 nm, 500 nm, 600 nm, and 700 nm. The three types of sea ice are quasi-infinite, and their properties are described in Table 1. These data demonstrate a large difference in how responsive each type of sea ice is to increasing mass ratios of *Romashkino* or *Petrobaltic* oil.



## 3.3 The effect of oil droplet size distribution on the albedo of quasi-infinite sea ice

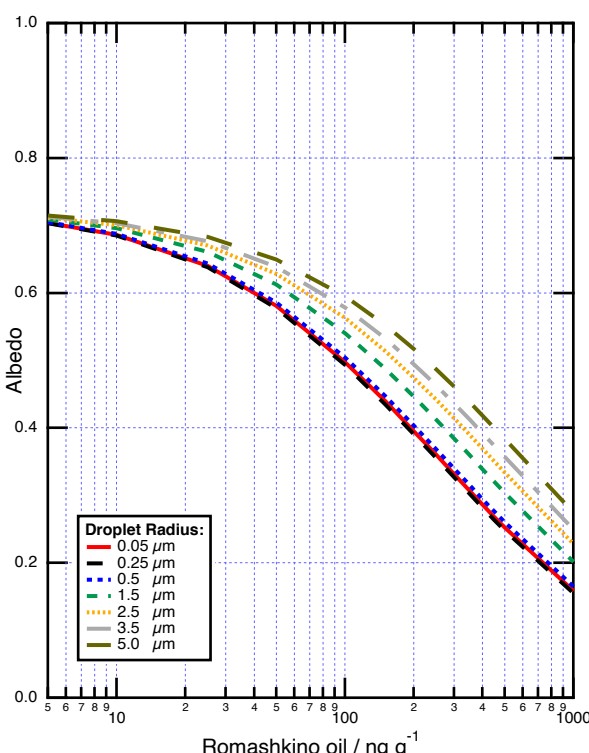

**Fig 3.** The albedo response of melting sea ice as mass ratios of *Romashkino* oil increase (0–1000 ng g$^{-1}$) at a wavelength of 400 nm. The different coloured lines represent different lognormally distributed droplet sizes with median radii of 0.05–5.0 μm. The melting sea ice is quasi-infinite, and the properties are described in Table 1. These data demonstrate the sea ice albedo to be most responsive to the smallest droplet sizes (0.05–0.5 μm).

Figure 3 illustrates the surface albedo response of melting sea ice at a wavelength of 400 nm as mass ratios of *Romashkino* oil increase from 0–1000 ng g$^{-1}$ for different sized spherical droplets. The different droplet sizes examined are characteristic of oils that have undergone different extents of weathering and are common in coastal areas, with radii of 0.05 μm, 0.25 μm, 0.5 μm, 1.5 μm, 2.5 μm, 3.5 μm, and 5 μm, in line with Haule and Freda (2016) and utilising the geometric standard deviation that were experimentally measured in Otremba (2007). The albedo is most responsive to the smallest droplets sizes (0.05–0.5 μm) and progressively less responsive as the droplets increase in size. The difference in albedo response to the smallest droplets is small, with a decrease in albedo at a mass ratio of 1000 ng g$^{-1}$ to 21.5% of the unpolluted value for a 0.25 μm droplet, 22% for a 0.05 μm droplet, and 22.8% for a 0.5 μm droplet. Figure 4 shows the mass absorption coefficient of a larger droplet size range (0.005–50 μm) of both *Romashkino* and *Petrobaltic* oils, which are assigned a geometric standard deviation of $e^1$ and are displayed at a wavelength of 400, 500, 600, and 700 nm. The mass absorption coefficient of oil droplets is a function of droplet size in the Mie regime and can be seen to peak for *Romashkino* oil at a wavelength





of 400 nm between 0.05 to 0.5 µm in Fig. 4. The relationship shown in Fig. 4 is an interplay between the value of the absorption efficiency from the Mie calculation and the increasing mass of a larger droplet, resulting in a decreasing mass absorption coefficient as the droplet size increases. As the wavelength
increases the critical radius (i.e. the radius with maximum mass absorption coefficient) for *Romashkino* also changes from 0.1 µm at 400 nm to 0.5 µm at 700 nm. Similarly, for *Petrobaltic* the critical radius changes with increasing wavelength from 1 µm at 400 nm to 5 µm at 700 nm. For *Romashkino* there is only a minor decrease in mass absorption coefficient for droplets smaller than the critical radius, with a 0.005 µm droplet decreasing to 87.9% of the value at 400 nm, and 80.2% at 700 nm. However, there is a
significant decrease in mass absorption coefficient for droplets larger than the critical radius, with a 50.0 µm droplet decreasing to 5.4% of the value at 400 nm, and 27.7% of the value at 700 nm. For *Petrobaltic* there is a comparable minor decrease in mass absorption coefficient for droplets smaller than the critical radius, with a 0.005 µm droplet decreasing to 78.6% of the value at 400 nm, and 46.2% at 700 nm. However, droplets larger than the critical radius do not experience the same reduction that is seen in the
*Romashkino,* and the value only decreases to 75.5% at 400 nm, and 90.8% at 700 nm for the radii considered, presumably owing to the significantly weaker mass absorption coefficient. Figure 4 demonstrates the need for the size distribution of oil to be know for accurate reproductions of oil spills to be possible.

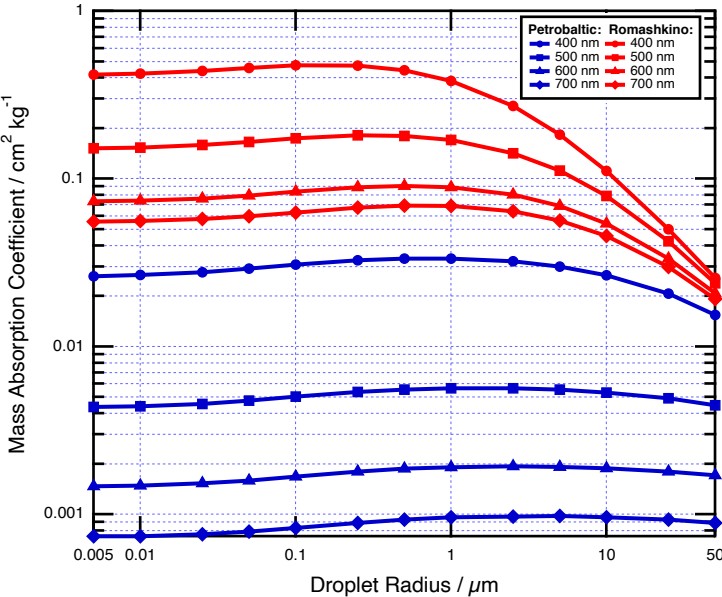

**Fig. 4.** Mass absorption coefficient of *Romashkino* and *Petrobaltic* oils with peak droplet radii between 0.005–50 µm. The geometric standard deviation size was set to $e^1$ to represent a midrange value for the particles studied here. These data demonstrate the mass absorption coefficient of *Romashkino* and *Petrobaltic* oil peak at different droplet size and at different wavelengths.



## 3.4 The effect of increasing background black carbon on the albedo of quasi-infinite sea ice

Figure 5 illustrates the surface albedo of melting, first-year, and multi-year sea ice as a function of increasing mass ratios of *Romashkino* oil from 0–1000 ng g$^{-1}$ and background black carbon levels from 1–100 ng g$^{-1}$. The upper row of Fig. 5 (A, C, and E) indicate albedo versus increasing mass ratios of *Romashkino* oil, whereas the lower row (B, D, and F) is a metric for the sensitivity of an ice to oil in the presence of black carbon, where sensitivity is the rate of change in albedo with increasing mass ratios of
*Romashkino* oil.
      Black carbon has a very strong absorption cross section over shortwave wavelengths which is approximately ~10 m$^2$ g$^{-1}$ between 400–700 nm (Marks and King, 2013), although global mean values range between 4.3–15 m$^2$ g$^{-1}$ (Bond et al., 2013). If black carbon is present the three types of sea ice have a decreased sensitivity to increasing mass ratios of oil pollution, as the largest decrease in albedo occurs
in the cleanest ices with the smallest black carbon mass ratios. At a 1 ng g$^{-1}$ mass ratio of black carbon, the albedos of the three sea ices are very responsive to low mass ratios of oil of less than 100 ng g$^{-1}$. Melting sea ice is the most responsive sea ice to oil and has the largest sensitivity at 1ng g$^{-1}$ of black carbon, whereas multi-year sea ice is the least responsive and has the lowest sensitivity comparatively. As the mass ratio of black carbon increases, the sensitivity of the three sea ices to increasing mass ratios
of oil rapidly decreases. Even at typical Arctic values of black carbon, which range between 5.5–19.2 ng g$^{-1}$, the sensitivity is significantly decreased (Grenfell et al., 2002; Jiao et al., 2014). The 100 ng g$^{-1}$ black carbon mass ratios are extreme values (Jiao et al., 2014) and significantly decrease the albedo of all three ices without any oil pollution present: multi-year sea ice albedo decreases to 58.9% of its 1 ng g$^{-1}$ black carbon value, the first-year sea ice albedo decreases to 32.1%, and the melting sea ice decreases to 11.5%.
At such high mass ratios of black carbon, the responsive of the sea ice is significantly decreased as mass ratios of oil increase. Therefore, oil pollution will have a significantly greater effect on the albedo of the cleanest ice that is not already polluted by large mass ratios of black carbon.



**Fig. 5.** Albedos of multi-year (A), first-year (C) and melting (E) sea ice with increasing mass ratios of
*Romashkino* oil (0–1000 ng g$^{-1}$) with a peak droplet radius of 0.05 μm. The gradients ($\frac{\Delta A}{\Delta m}$) of plots A, C,
and E are shown for multi-year (B), first-year (D), and melting (F) sea ice, respectively. The selected
wavelength is 400 nm and the background levels of black carbon increase: 1 ng g$^{-1}$ (black line), 10 ng g$^{-1}$ (red line), 20 ng g$^{-1}$ (blue line), and 100 ng g$^{-1}$ (green line). The three types of sea ice are quasi-infinite,
and their properties are described in Table 1. These data demonstrate that increasing mass ratios of black
carbon rapidly decrease the albedo response of each type of sea ice to *Romashkino* oil.

## 3.5 The effect of oil, type of sea ice, and mass ratio on realistic thickness sea ice albedo

Calculations in section 3.2 are replicated with realistic thickness melting, first-year, and multi-
year sea ice as a function of increasing mass ratios of *Romashkino* and *Petrobaltic* oils from 0–1000 ng
g$^{-1}$ and are illustrated in Fig. 6. Typical Arctic thicknesses of 2.5 m for multi-year and melting sea ice
(Bourke and Garrett, 1987; Laxon et al., 2003; Weeks, 2010), and 0.8 m for first-year sea ice (Weeks,
2010) are used with a realistic background Arctic black carbon mass ratio of 5.5 ng g$^{-1}$ (Grenfell et al.,
2002). The realistic thickness sea ices have different unpolluted albedos from the quasi-infinite sea ice





(melting sea ice 0.41, first-year sea ice 0.57, and multi-year sea ice 0.87 at a wavelength of 400 nm respectively) and are comparable with previous literature (Marks and King, 2014; Lamare et al., 2016).

The albedo response of each sea ice is decreased from the quasi-infinite sea ice albedo response owing to the presence of the dark ocean layer below the ice, with a 1000 ng g$^{-1}$ mass ratio of *Romashkino* oil resulting in the albedo decreasing to 75.3% for the multi-year sea ice; 66.7% for the first-year sea ice; and 35.7% for the melting sea ice relative to the unpolluted albedo at a wavelength of 400 nm. The albedo response to increasing mass ratios of *Romashkino* oil at longer wavelengths is weaker and essentially

insignificant at wavelengths longer than 500 nm. Similarly, the albedo response to *Petrobaltic* oil at a 1000 ng g$^{-1}$ mass ratio is negligible, with albedo decreasing to 96.9% for the multi-year sea ice; 97% for the first-year sea ice; and 90.9% for the melting sea ice relative to the unpolluted albedo at a wavelength of 400 nm. Evidently the thickness and type of sea ice, as well as the type of oil, are important components in the effect that increasing mass ratios of oil have on sea ice albedo. As the ice becomes thinner photons

are likely to penetrate the ice and be absorbed by the underlying dark ocean layer. In the quasi-infinite sea ice photons were only weakly absorbed by the ice and efficiently by the oil. The first-year sea ice is relatively less responsive to increasing mass ratios of oil than either the melting sea ice or multi-year sea ice owing to its 0.8 m thickness compared to a 2.5 m thickness. This decreased response in albedo is indicated by the first-year sea ice albedo decreasing by 0.19 as the *Romashkino* oil mass ratio increases

from 0 to 1000 ng g$^{-1}$, whereas the multi-year sea ice albedo decreases by 0.21. The dramatic difference in albedo response to *Romashkino* oil and *Petrobaltic* oil indicates that the type of oil has a greater effect on sea ice albedo than the type or thickness of sea ice.





**Fig. 6.** Albedos of multi-year, first-year, and melting sea ice with increasing mass ratios of *Romashkino* (A, B, and C) and *Petrobaltic* (D, E, and F) oils (0–1000 ng g$^{-1}$) with a median droplet radius of 0.05 μm. The selected wavelengths are 400 nm, 500 nm, 600 nm, and 700 nm. The melting sea ice and multi-year sea ice are 2.5 m thick; the first-year sea ice is 0.8 m thick, and their properties are described in Table 1. These data demonstrate that the type of oil rather than the type of sea ice may have the largest effect on the albedo of realistic thickness sea ice.



# 4 Discussion

The discussion is separated into seven sections: the Mie calculations, the effect of different droplet size distributions, the response of sea ice albedo to *Petrobaltic* or *Romashkino* oil, the effect of *Romashkino* and *Petrobaltic oil* on different types of sea ice, the albedo response of quasi-infinite sea ice
and realistic thickness sea ice to oil pollution, potential implications of the study, and potential limitations of the study.

## 4.1 Mie calculations

Several studies have explored the effects of aerosol pollution on snow and sea ice (e.g. Warren and Wiscombe, 1980; Warren, 1984; Clarke and Noone, 1985; Warren and Clarke, 1990; Light et al.,
1998; Grenfell et al., 2002; Aoki et al., 2003; Lee-Taylor and Madronich, 2002; Jacobson, 2004; Flanner et al., 2007; Zender et al., 2009; Doherty et al., 2010; Dumont et al., 2010; Zatko et al., 2013; Marks and King, 2013; 2014; Lamare et al., 2016; Marks et al., 2017) but there has been limited research into the effect of oil pollution other than two basic field studies by NORCOR (1975) and Gavrilo and Tarashkevich (1992). To the authors knowledge, this is the first time the effect that small mass ratios of
dispersed submicron weathered oil droplets have on the wavelength dependent albedo of sea ice are considered. Whilst studies have looked at how oil pollution affects the inherent optical properties (i.e. absorption, scattering, backscattering, and attenuation coefficients) of liquid seawater (e.g. Otremba, 2007; Haule et al., 2015; Haule and Freda, 2016), they have not examined what the effect is on sea ice and its albedo. In this study, background levels of oil pollution are considered over a wide area and not
within close proximity to any single large and acute event (e.g. Macondo Oil Spill or Exxon Valdez).

Oil is known to entrain itself into sea ice in several ways: (1) oil beneath the sea ice is frozen into the sea ice and the ice sheet continues to grow underneath the sea ice, with the oil being fully encapsulated in the ice matrix within 18–72 hours (NORCOR, 1975; Dickins and Buist, 1981; Buist and Dickins, 1983; Buist et al., 1983; Wilkinson et al., 2015); (2) Oil will drift with the sea ice in broken ice fields when
concentrations of sea ice are greater than 60–70% and is incorporated into the sea ice during freeze up (Venkatesh et al., 1990); (3) the oil can become trapped in small cavities under the sea ice and may or may not be encapsulated, depending on the time of year (Drozdowski et al., 2011). Once encapsulated in the sea ice the oil can migrate to the surface owing to its buoyancy, however this rate is heavily dependent upon the amount of brine drainage within the sea ice and therefore the time of year (Drozdowski et al.,
2011). Martin (1979) found that during the winter oil formed thin lenses in the sea ice, which then distributed themselves throughout the brine channel feeder systems in the spring (February to March) as these expanded, and eventually formed horizontal layers in the upper part of the sea ice (NORCOR, 1975). These studies have focused on macroscopic quantities of oil whereas the work presented here focuses on microscopic sized background concentrations of oil which require more work. Therefore, it is unclear
whether the droplets are in the matrix or brine channels of the sea ice, so in the work presented here they are assumed to be evenly distributed throughout the ice to replicate the different stages of the seasonal variation. The droplets are also considered to be in a pure sea ice medium (Warren and Brandt, 2008) as opposed to a brine medium (Quan and Fry, 1995) for optical calculations. The refractive index of the medium in which the oil droplet is contained may effect the value of its mass absorption coefficient





established from a Mie calculation. However, as indicated by Fig. 7, the difference in mass absorption coefficient for the oil droplets between a sea ice medium (refractive index of 1.3091 at a wavelength of 600 nm) and brine (refractive index of 1.3402 at a wavelength of 600 nm) medium are inconsequential owing to their similar refractive indices (Quan and Fry, 1995; Warren and Brandt, 2008). Thus, the position of the oil droplet within the sea ice may not be important.

The shape of the oil droplets may also be an important factor that influences the Mie calculations. Whilst the shape of oil droplets in sea ice or brine is not known, immiscible fluids are known to adopt spherical morphologies to reduce their surface relative to volume and both mechanically and chemically dispersed droplets are spherical in nature (Johansen et al., 2013). Brine channels typically have a width of 0.1 mm (Lieb-Lappen et al., 2017) which is far larger than the droplet sizes considered here, so it is

unlikely that droplet morphology is affected when being transported through the brine channels (NORCOR, 1975; Martin, 1979). Therefore, the oil droplets are assumed to be spherical in the model as spheres are computationally facile for Mie calculations and the shape must be known for the calculation.

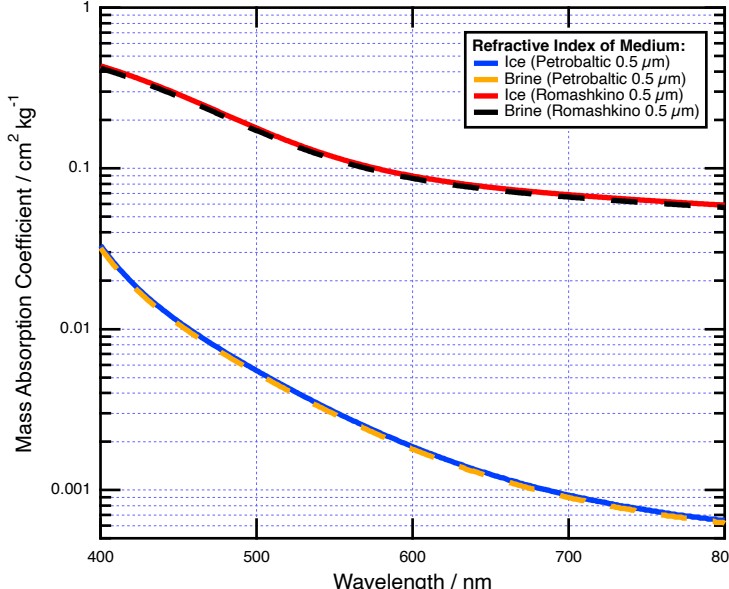

**Fig. 7.** Mass absorption coefficient of *Romashkino* and *Petrobaltic* oils with a median droplet radius of
0.5 µm in a medium of pure sea ice (Warren and Brandt, 2008) and a medium of brine (Quan and Fry, 1995). These data demonstrate that the mass absorption coefficient is independent of location (i.e. within the brine channel or ice matrix).

### 4.2 Droplet size distribution

Tens of microns to millimetre scale droplets are present in seawater immediately after the release
of oil, from either the subsurface or at the surface (e.g. Spaulding et al., 1992; Reed and Rye, 1995; Daling et al., 1997; Johansen, 2003; Zheng et al., 2003; Lima Neto et al., 2008; Socolofsky et al., 2008; Gamzaev, 2009; Papadimitrakis et al., 2005, 2011; Fraga et al., 2016; Dissanayke et al., 2018). Fingas and Hollebone (2014) summarise much of the literature on the behaviour of oil in ice-infested waters, however all the



research discussed involves releasing large quantities of oil either directly under or onto the ice in acute
events, whereas this study focuses on background quantities of oil. After release the droplets can travel
several hundred square kilometres within several days (Berenshtein et al., 2020) and are exposed to
weathering processes that continually decrease the droplet size (e.g. Daling et al., 1990; Resby and Wang,
2004; Dillipiane et al., 2021). The dispersed droplet distribution sizes can be described by a lognormal
distribution of submicron droplets with an experimentally measured peak for weathered oil centred
around a median radius of 0.05 µm (Otremba, 2007; Z. Li et al., 2011; Johansen et al., 2013; Haule et al.,
2015; Haule and Freda, 2016). The focus of this study is on the small mass ratios (i.e. less than 1000 ng
g$^{-1}$) of submicron weathered oil droplets that are incorporated into the sea ice in polluted polar seas. It is
likely that small droplets in this range are the most numerous and will affect large areas of polar sea ice,
in addition to the fact that droplets that have been treated with dispersants will also be smaller (C. Li et
al., 2017). Therefore, lognormal distributions of droplets with medians in the range 0.005–50 µm are
considered for completeness (Fig. 4). The droplets considered in Fig. 4 were assigned a geometric
standard deviation size $e^1$ as it is a mid-range value from Haule and Freda (2016) and to keep the
distribution width equivalent for each droplet size. The geometric standard deviation used for all other
figures is consistent with measurements from Otremba (2007) and used in Haule and Freda (2016).

555         With reference to Fig. 3, it is apparent that sea ice albedo is weakly sensitive to the droplet size of
oil in the range explored. Weathered submicron droplets of absorbing oils such as *Romashkino* have a
greater effect on albedo than larger micron-scale droplets, with Fig. 4 indicating the critical radius changes
marginally from 0.1 µm at a wavelength of 400 nm to 0.5 µm at a wavelength of 700 nm. These weathered
submicron droplets are roughly equivalent in size to the wavelength of light (~0.5 µm) and therefore Mie
light scattering must be considered (Petty, 2006). The *Petrobaltic* oil mass absorption coefficient also
decreases as the droplet size increases, albeit more slowly than the *Romashkino* oil owing to its very low
values. This weak dependence of sea ice albedo to droplet size may have implications for remote sensing
and droplet size estimates may be important to calculate the effect of oil pollution on sea ice albedo for
future events.

**4.3 Effect of different oils in quasi-infinite sea ice**

        Both the real and imaginary refractive indices of *Romashkino* and *Petrobaltic* oil are described as
the respective largest and smallest values that oils capable of creating surface films possess, according to
Otremba (2000). Therefore, *Romashkino* have a significantly larger mass absorption cross section than
*Petrobaltic*; roughly 13-fold greater for a 0.5 µm droplet at a wavelength of 400 nm. The resulting
differences in effect on sea ice albedo are therefore very significant, with 1000 ng g$^{-1}$ of *Romashkino*
decreasing albedo of first-year ice from 0.87 to 0.41 (a decrease to 47% relative to the unpolluted value)
at 400 nm, whereas *Petrobaltic* decreases the albedo from 0.87 to 0.77 (a decrease to 88.2% relative to
the unpolluted value). As the ice becomes more absorbing at longer wavelengths (Warren and Brandt,
2008) this difference is decreased: at 550 nm 1000 ng g$^{-1}$ of *Romashkino* causes a decrease in albedo to
81.6% and *Petrobaltic* to 99.4%. At 700 nm the effect is very low for both oils. Thus, even at small mass
ratios, heavy, and opaque oils such as *Romashkino* (0.94 g cm$^3$) will have a significant effect on sea ice
albedo whilst relatively transparent and light oils, such as *Petrobaltic* (0.77 g cm$^3$), may have an
insignificant effect on albedo (Otremba, 2005; 2007; Hollebone, 2015). Therefore, both *Romashkino* and



*Petrobaltic* can be regarded as the upper and lower respective brackets of the effect that oil pollution can
have on sea ice albedo. This may be of note as the most commonly used fuel for commercial vessels is
Heavy Fuel Oil which is defined as having a density of 0.9 g cm$^{-3}$ (IMO, MARPOL Annex I) and is
therefore analogous to *Romashkino* oil.

**4.4 Response of melting, first-year, and multi-year sea ice to oil**

The three different sea ices considered here are all modelled at a quasi-infinite thickness of 15 m,
in line with similar studies (e.g. Marks and King, 2014; Lamare et al., 2016) to ensure that the albedo was
independent of the layer under the sea ice and any effects that a non-optically thick sea ice may have,
whilst the properties of sea ice type, mass ratio and droplet size are considered. Although such large
thicknesses are not typical in nature and this causes the unpolluted albedo of all three sea ices to be
exaggerated from their natural values, it is useful to understand how the Romashkino and Petrobaltic
fundamentally affect albedo. The multi-year sea ice is the least affected by either type of oil, with a
decrease to 70.9% and 94.5% of the unpolluted albedo at a mass ratio of 1000 ng g$^{-1}$ of *Romashkino* oil
and *Petrobaltic* oil, respectively. The multi-year sea albedo ice is only affected by mass ratios of
*Romashkino* oil exceeding 100 ng g$^{-1}$. However, multi-year sea ice is susceptible to oil pollution as
Comfort and Purves (1982) confirmed that oil does eventually migrate to the surface, despite a thickness
of several metres. First-year sea ice is more sensitive to oil pollution and decreased to 47.1% and 88.2%
of the unpolluted albedo at a mass ratio of 1000 ng g$^{-1}$ of *Romashkino* oil and *Petrobaltic* oil, respectively.
The first-year sea ice albedo is affected by mass ratios of *Romashkino* exceeding 50 ng g$^{-1}$. The most
sensitive sea ice to oil pollution by a considerable margin is the melting sea ice which decreased to 22%
and 77% of the unpolluted albedo at a mass ratio of 1000 ng g$^{-1}$ of *Romashkino* oil and *Petrobaltic* oil,
respectively. The melting sea ice albedo is affected by mass ratios of *Romashkino* exceeding 25 ng g$^{-1}$.

Melting sea ice is the least reflective type of sea ice with a scattering cross section of 0.03
compared to 0.15 (first-year sea ice) and 0.75 (multi-year sea ice) (Perovich, 1996; Marks and King,
2014), resulting in it becoming substantially more absorbing as pollutants are added. Whilst these findings
are unsurprising, they are concerning as multi-year Arctic sea ice has declined dramatically from 64% of
the total sea ice in March 1985 to 30% in March 2020, with 4+ year old sea ice decreasing from 33% to
4.4% respectively (Perovich et al., 2020). As these data shows, this decline in perennial types of sea ice
renders the Arctic much more vulnerable to increased oil pollution in the region, particularly as it is
opened to both shipping and oil extraction.

**4.5 The effect of depth**

The work described here has focused on quasi-infinite sea ice throughout to ensure that it is
optically thick and therefore independent of the layer under the sea ice. Section 4.5 explores the response
of realistic sea ice thicknesses and black carbon mass ratios to offer a realistic comparison with the quasi-
infinite thicknesses. In comparing the quasi-infinite sea ice with the realistic sea ice (i.e. Fig. 2 and Fig.
6) it is apparent that the melting sea ice is less responsive to increasing mass ratios of Romashkino oil
and essentially not affected by *Petrobaltic* oil. The photons are being continuously scattered in the quasi-
infinite sea ice and this increases the likelihood of the photons being absorbed by oil droplets, whereas in



the realistic ice the photons have a greater chance of passing through the ice and being absorbed by the dark layer under the sea ice. This is also apparent in the first-year sea ice which is less affected by increasing mass ratios of both oils than the multi-year sea ice, despite having a smaller scattering cross
section, because it is thinner. Even though the realistic types of sea ice are less responsive to increasing mass ratios of oil, their albedos are still significantly decreased by the *Romashkino* oil. Therefore, it appears that the type of oil has the biggest effect on how responsive sea ice is to increasing mass ratios of oil as opposed to the type of sea ice and in contrast with the findings of a comparable study into the effects of mineral dust on sea ice albedo (Lamare et al., 2016).

## 4.6 Implications of study

Arctic multi-year and first-year sea ice are declining at 17.5% and 13.5% respectively and melting sea ice is becoming more prevalent earlier in the year (Comiso, 2012; Tschudi et al., 2019). First-year and particularly melting sea ice are more responsive to oil pollution than multi-year sea ice, so these trends indicate that sea ice albedo in the Arctic may become more vulnerable to background levels of oil
pollution as the ice becomes progressively thinner and younger. This increasing vulnerability will be exacerbated by potential future extraction of the vast hydrocarbon reserves in the Arctic Ocean (Bird et al., 2008; Krivorotov and Finger, 2019; Czarny, 2019) and the development of northern sea routes (Ho, 2010; Eguíluz et al., 2016; Kikkas and Romashkina, 2018), resulting in vast quantities of Heavy Fuel Oil akin to *Romashkino* oil passing through the region and potentially exposing it to increased pollution (De
Carolis et al., 2014; IMO, MARPOL Annex 1). Furthermore, the albedo of clean sea ice is more responsive to oil pollution than sea ice polluted with higher mass ratios of aerosols such as black carbon, so future oil pollution may imperil vast areas of pristine Arctic sea ice even at low background mass ratios.

## 4.7 Uncertainties

There are a few potential uncertainties to this study stemming primarily from the fact that it is a modelling study and that, to the authors knowledge, this is the first instance that the climatic effect of oil pollution on sea ice has been considered. A limitation with the Mie calculations is the uncertainty of whether the oil droplets are spherical in the ice. A careful study of the particle morphology of oil droplets in ice is required to improve the veracity of future modelling studies. However, it is expected that the
results presented here would be much better than an order of magnitude level. A second potential limitation of this study is that it has not considered the effect of a snowpack on top of the sea ice. Whereas similar studies (e.g. Marks and King, 2013, 2014; Lamare et al., 2016) consider layers of snow on top of the ice, they differ in that the pollutants explored are atmospherically deposited directly on top of both the ice and snow. For the oil considered here to be incorporated into the ice it must be frozen into the ice
directly from the seawater below (NORCOR, 1975; Martin, 1979; Dickins and Buist, 1981; Buist and Dickins, 1983; Buist et al., 1983; Wilkinson et al., 2015; Drozdowski et al., 2011). However, Owens et al. (2005) found that if snow comes into contact with oil, for instance if the snow-ice forms from the surface flooding of an ice floe, it can initially absorb up to 70% of the oil thus decreasing the albedo of the snow. Furthermore, a layer of oil underneath the snowpack may expedite melting from below as the





oil may be up to 6 ºC warmer than the air temperature (Chen, 1972). The amount of snow cover on sea ice varies significantly across the Arctic (Zhou et al., 2021) so some areas may be more vulnerable to oil pollution earlier in the spring. Nevertheless, even a thin snow cover (less than 1 cm) can increase the albedo of sea ice and can be considered optically thick when it exceeds 10 cm (France et al., 2011; Marks and King, 2013). Therefore, similar to the work by Light et al (1989), the findings of this study may only

be valid during the ablation season when snow cover has melted or been removed by wind.

## 5 Conclusion

Background mass ratios of crude oil ($\leq$1000 ng g$^{-1}$) can have a significant effect on sea ice albedo. The albedo response is dependent on the type of oil, with the relatively light absorbing *Romashkino* oil having the largest effect owing to its large mass absorption coefficient. The albedo response is also

dependent on the type of sea ice, with a 1000 ng g$^{-1}$ mass ratio of *Romashkino* oil resulting in quasi-infinite thickness (15 m thick) multi-year sea ice decreasing to 70.9%, first-year sea ice decreasing to 47.9%, and melting sea ice decreasing to 22% relative to the unpolluted albedo at a wavelength of 400 nm. Sea ice thickness also plays an important role on the albedo response with realistic thickness multi-year sea ice decreasing to 75.3%, first-year sea ice decreasing to 66.7%, and melting sea ice decreasing

to 35.7% of the unpolluted value. The size of the oil droplets has a weak effect on the albedo response, with weathered submicron *Romashkino* oil droplets (0.05–0.5 µm) being the most absorbing in the visible spectrum (400–700 nm). This work demonstrates that the type of oil, type of ice, thickness of ice, and droplet size are important components in the response of sea ice albedo to background mass ratios of crude oil pollution. The type of oil is the most important factor in how responsive sea ice is to increasing

mass ratios of oil and is more important than the type of sea ice being polluted. By collecting these data in the event of a future spill, it may be possible to model what the effect of the background oil pollution may be. This may become increasingly important as multi-year sea ice continues to decrease, first-year sea ice becomes more dominant and there is earlier melting of sea ice in the year.

*Acknowledgements*. The work contained in this paper contains work conducted during a PhD study

undertaken as part of the Centre for Doctoral Training (CDT) in Geoscience and the Low Carbon Energy Transition. It is sponsored by Royal Holloway, University of London via their GeoNetZero CDT Studentship whose support is gratefully acknowledged.

Edited by:

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
