# Peer review of "Quantifying the effects of background concentrations of crude oil pollution on sea ice albedo"

_The Cryosphere, 2021_

## Author Comment (AC2)

**Response to Anonymous Referee #2 on tc-2021-372**

1. The authors would like to thank the reviewer for their positive comments and the recommendation that this manuscript should be published.

2. Reviewer comment: '*There is a misalignment between the title of the article and certain statements (see detailed comments below) and the actual purpose of the proposed work. Indeed, the climatic impact of crude oil pollution is not addressed in this sensitivity study. I would recommend that the authors rephrase the title of the manuscript and correct the statement line 641.*'

- Author response: The title has been changed to '*Quantifying the effects of background concentrations of crude oil pollution on sea ice albedo.*'

- The reviewer raises a valid point, the full climatic potential has not been realised by our paper as we only explore albedo. We have therefore removed the word climatic from line 641.

3. Reviewer comment: '*The impacts of crude oil droplets dispersed in sea ice are investigated in the visible wavelengths (400-700 nm). Could the authors please expand on the reasons for this wavelength range? Is it a limitation of the model or a deliberate choice? Most observation tools (ground-based instruments, drone or aircraft mounted sensors, Earth Observation satellite) cover a larger spectrum, generally from the visible to the shortwave infra-red. Furthermore, climate models (e.g. CMIP models) consider the longwave radiative balance to monitor the Earth's energy balance. Although longwave radiation is most likely out of scope for this study, it is surprising that such a small range of wavelengths is being considered here. Extending the range to 2500 nm would allow direct comparisons with observations.*'

- Author response: The selected wavelengths are a deliberate choice by the authors as we are only measuring over the wavelength window where oil has an effect on the albedo of sea ice.

- To clarify this for the reviewer we have attached a figure below comparing the absorption coefficients of oil taken from Warren and Brandt (2008) to background levels of Romashkino oil between 10–1000 ng g$^{-1}$:

[Figure]

-
- **Fig. A1.** Wavelength dependent absorption coefficients of ice measured by Warren and Brandt (2008) and Romashkino oil at increasing background mass ratios (10–1000 ng g-1).
- At a wavelength of 580 nm the absorption coefficient of a mass ratio of 1000 ng g$^{-1}$ of Romashkino oil is equal to the absorption of ice. At higher wavelengths the absorption of ice is greater than that of the oil. At lower mass ratios of oil, the absorption coefficient is equal to ice at shorter wavelengths.
- This effect is illustrated in Fig. 1 of the paper where oil can be seen to have a decreasing effect on albedo as the wavelength increases. Similarly, the peak in albedo moves to shorter wavelengths as mass ratios of oil decrease as the absorption coefficient of oil decreases with increasing wavelength in the range explored here (400–700 nm) whilst the absorption coefficient of ice increases at longer wavelengths.
- At 700 nm the absorption of solar photons is dominated by ice as shown by the intersection in Fig. A1, so longer wavelengths are not considered in the present study.

- The reviewer is correct to point out that remote sensing tools measure in the infrared range of the spectrum, but we hope that Fig. A1 coupled with Fig. 1 in the paper show that the absorption in this region is mostly due to the ice, not due to added to oil. Therefore, we have not explored further than 700 nm as the albedo at longer wavelengths are controlled by the absorption of the ice.
- For clarity in the text, we have edited the following line in Section 2.1: *'This study considers the wavelength range 400–700 nm as the absorption coefficient of oil decreases with increasing wavelength whereas the absorption coefficient of ice increases with wavelength, thus at wavelengths longer than 700 nm the absorption of solar photons is dominated by ice.'*

4. Reviewer comment: *'The authors state that "the effects that oil pollution has upon sea ice albedo have not previously been considered in literature". Although this statement holds, the authors disregard the existing corpus of works that investigate the effects of oil pollution on sea ice reflectance (e.g. 1-5 in the reference section below). Despite the quantities being different, albedo can be derived from reflectance using a BRDF model, and it is widely accepted that reflectance may be used as an approximation for albedo. A short review of the existing studies would be desirable in the introduction.'*
- Author response: The reviewer is correct to mention albedo can be calculated from BRDF and we would like to thank them for highlighting these papers which are new to the paper (barring [2] Liu et al (2018)), and these have now been included in the paper.
- We have now edited the following line in section 1: *'studies have looked into the hyperspectral features of oil-polluted sea ice (Praks et al, 2004; Ivanov et al, 2005; Liu et al, 2016, 2018; Chao et al., 2017); it is possible to calculate the albedo from bidirectional reflection distribution function (BDRF), however these are not comparable to this study as background mass ratios of oil are examined here.'*

5. Reviewer comment: *'In this study the authors have chosen to distribute the oil evenly throughout the sea ice. While this may be realistic in certain conditions (particularly for low oil concentrations), how plausible is this to occur at higher loadings (1000 ng g-1)? In the discussion (line 496) the authors describe the different scenarios of how oil entrains itself into sea ice. From these comments, it is clear that layering of the oil is a common situation encountered in sea ice. The model used allows the definition of layers throughout the ice pack: rather than address the relationship between black carbon and oil loadings, would it not have been of value to consider the effects of oil located in specific (e.g. surface, or subsurface) layers?.'*
- Author response: The reviewer is correct to mention that the model can do layers, however, in previous studies where layering was explored (e.g. Marks and King, 2013; Lamare et al, 2016; Marks et al, 2017) this was owing to the fact that there was sufficient prior knowledge of how aerosols (e.g. black carbon/dust) deposited from above are found in snow and ice. To accurately model these layers, we require more experimental data about how background mass ratios of oil are incorporated in layers in the ice before it is feasible to model it. It is also likely that unlike aerosol layers in the snow or ice, an oil layer would not be static. The parameter space needed to consider the number, thickness, depth of all these layers and thickness of

three different types of realistic and quasi-infinite ice is huge. To cover this parameter space (wavelength dependence, droplet size, mass ratio of droplets, ice thickness, and position of layer) would be a much larger study requiring further knowledge of the behaviour of background oil within ice and which would benefit from an associated field or lab-based study.

- We have added the following text to Section 1: '*The TUV-snow model is capable of considering layers of pollutant in the ice; however, until there is greater knowledge of the thickness, type and location of these layers for background mass ratios of oil, it is beyond the scope of the present work.*'

6. Reviewer comment: '*In link to the paragraph above, the concentrations of the oil would deserve more clarifications. Indeed, the article focusses on "microscopic sized background concentrations of oil" (line 509) but in the introduction it is mentioned that after the Deepwater Horizon incident, mass ratios of 100 ng g-1 were found. Can mass ratios of 1000 ng g-1 still be considered as "background"?.*'

- Author response: The authors mention the DWH mass ratio of 100 ng $g^{-1}$ as this was tracked in Berenshtein et al (2020) to travel from the Mississippi Canyon to the Carolinas via the Loop Current, a distance exceeding 1500 km, indicating concentrations of oil can remain high over a very long range.
- Oil concentrations from several ppb to ppm are common in regions of intense shipping, marine transportation, and offshore oilfields (Haule and Feda, 2016) and other studies have examined the optical properties of oil in the marine realm at concentrations of 1 ppm (i.e. 1000 ng $g^{-1}$) (Otremba, 2007).
- Whilst 1000 ng $g^{-1}$ may be considered high for background oil concentrations for areas of the Arctic Ocean that are currently pristine, they are valuable for straightforward parameterisation of this study which the reviewer has described as of value to the scientific community.
- We have added the following line to Section 4.2: '*The mass ratio of 1000 ng $g^{-1}$ of oil in ice is possibly a large value for diffuse background pollution but is included to provide a significant upper limit to the effect of background oil pollution on sea ice.*'

7. Reviewer comment: '*More information about the relationship between the oil droplet size and the mass ratios would be important to better understand which scenarios in the paper are most plausible. Have the authors investigated if there is a relationship between droplet size and mass ratios between 1 and 1000 ng g-1 or is it likely to find all sizes within the loading range?*'

- Author response: The authors could not find the requisite answer to this question. In this study we have explored a large envelope of a realistic range of droplet sizes to cover all reasonable eventualities. Some of the high mass ratios with large sizes may be unlikely, however we have included them here for completeness and to provide an upper limit.
- Moreover, there is a lack of a large corpus of results for the smaller sizes, as is highlighted in and described extensively in Section 1 (L 128-147)

8. Reviewer comment: '*Section 4.6 on the implications of the study is quite light and could be fleshed out more. It would be insightful to read the authors thoughts on the implications in terms of how the sea ice melting rates and extent in summer will be*

*affected by oil pollution. How does the increased oil pollution impact the energy balance of the Arctic, and are the effects sufficient to be considered in General Climate Models?'*

- Author response: The authors have been asked to remove the climatic implications from both our title and text by both reviewers and we therefore feel it inappropriate to make climatic predictions.

9. Reviewer comment: *'Lastly, as a side note, I would suggest that the authors make the input data available through an open repository, which would benefit the modelling community greatly (e.g. use of the parametrisations of crude oil in climate models) and allow for further intercomparisons of modelling approaches.'*

- Author response: All data will be made available on Zenodo once the manuscript is accepted for publication.
- The following text has been added after Section 5: *'Data availability. All data have been published using Zenodo and can be accessed at: DOI.10.5281/zenodo.6514952.'*

**Detailed comments**

10. Reviewer comment: *'l46: What do the authors mean by: "The wavelength integrated and spectral albedos for different types of sea ice have previously been considered [...]: this study focuses on three types of sea ice: melting, first-year, and multi-year sea ice." It is not clear if the authors are referring to the literature or if they are stating that they have considered a wide variety of sea ice types before settling for the three mentioned.'*

- Author response: The authors are referring to the literature where the values used in the study have been selected from.
- To make it clearer for the reviewer and the reader, we have changed the text to the following: *'The wavelength integrated and spectral albedos for different types of sea ice have previously been considered in literature (Grenfell and Maykut, 1977; Perovich, 1990; Timco and Frederking, 1996; Perovich, 1996; Gerland et al, 1999); utilising these optical properties this study focuses on three types of sea ice: melting, first-year, and multi-year sea ice.'*

11. Reviewer comment: *'Table 1: One would expect the density of first-year, multi-year and melting sea ice to be different owing to differences in structure (brine channels, air bubbles...). The reference cited by the authors [Marks and King, 2014] states that the density of sea ice ranges 700–950 kg m-3. How do the authors justify the same fixed value for all sea ice types?'*

- Author response: The reviewer is correct, typical densities of the three ices tend to range between 700–950 kg m$^{-3}$ (Grenfell and Maykut, 1977; Perovich, 1990; Timco and Frederking, 1996; Perovich, 1996; Gerland et al, 1999), however, these have been approximated to 800 kg m$^{-3}$ to be directly comparable to Lamare et al (2016).
- Variation in scattering cross section caused by natural variation of density is much smaller than the natural variation in scattering cross section as shown in the new Fig. 8, produced to answer a question from Reviewer 1 and for the paper, and presented again below. The following line has been added to Section 4.7 to demonstrate this: *'There is a natural variation in sea ice density between 700–950 kg m$^{-3}$ (Grenfell and*

> *Maykut, 1977; Perovich, 1990; Timco and Frederking, 1996; Perovich, 1996; Gerland et al, 1999) which can be propagated to a variation in scattering cross section of approximately a factor of 0.88 and 1.19 of the original values for the lowest and highest reported densities, respectively. These ranges are much smaller than the natural variation in the scattering of sea ice, as shown in Fig. 8. The variation due to density of ice in this study is not considered important.'*

- The variation in density is small and would only affect the scattering of the light in our calculations. The authors would like to stress that the change in scattering cross section of the ice types, the thickness of the ice, the changing concentration of oils, the different types of oil, and the different droplet size are the fundamental controllers of albedo in this study.
- The effect of a very low (700 kg m$^{-3}$) or large (950 kg m$^{-3}$) value for density can be estimated because a change in density of 700/800 would be the equivalent to an error in scattering cross section of approximately 0.88, and 950/800 would be approximately 1.19. This is within the error limit that has been presented for Reviewer 1 and added to the paper as Fig. 8 indicating that the effect is much smaller than the natural variation in scattering due to other conditions, so we do not need to consider it in this study.
- To make this clear to the reader, the following line has been added to Section 2.1: *'The density of sea ice has been fixed in this study to be comparable to previous work (Lamare et al, 2016), however sea ice density can range between extremes of 700–950 kg m$^{-3}$ (Grenfell and Maykut, 1977; Perovich, 1990; Timco and Frederking, 1996; Perovich, 1996; Gerland et al, 1999).'*

[Figure]

**Fig. 8.** Wavelength dependent albedo of multi-year (A), first-year (B), and melting sea ice (C). Shown here is the variability in albedo for each type of sea ice based on scattering cross sections described in literature (Grenfell and Maykut, 1977; Perovich, 1990; Timco and Frederking, 1996; Perovich, 1996; Gerland et al, 1999), ranging from 0.5–1 m$^2$ kg$^{-1}$ for multi-year sea ice; 0.1–2 m$^2$ kg$^{-1}$ for first-year sea ice; and 0.01–0.05 m$^2$ kg$^{-1}$ for melting sea ice. Shown in red are the data with no oil pollution present and shown in blue are the data with 1000 ng g$^{-1}$ of *Romashkino* oil present. The thicker lines show the values of typical ice used in this study. The melting sea ice and multi-year sea ice are 2.5 m thick; the first-year sea ice is 0.8 m thick, and the background concentration of black carbon is set to 5.5 ng g$^{-1}$.

12. Reviewer comment: 'Table 2: On what basis were the number of layers (201) and the increments chosen for the model?'
- Author response: Previous works (Marks, 2017) have demonstrated that a large number of layers are required at the interface between air and ice, and 201 layers was a good compromise between computational time and precision. Running the

model with fewer and more layers gave the same results. Previous calculations have increased the number of layers to give a constant answer.

- The following text has been added to the Methodology: '*Marks (2017) found it important to have a large number of layers at the interface so 201 layers offers a good compromise between computational time and precision.*'

13. Reviewer comment: '*l236: In this paragraph the authors describe the optical properties of the two types of crude oil used in the study. Although it is stated that "Whilst both crude oils have a variety of uses, including as marine engine fuels, Romashkino can be considered a typical marine engine Heavy Fuel Oil", the reasons for selecting Romashkino and Petrobaltic oil is not sufficiently clear to the reader, who has to wait until line 578 to understand that "both Romashkino and Petrobaltic can be regarded as the upper and lower respective brackets of the effect that oil pollution can have on sea ice albedo". Furthermore, it would be useful to understand if these oil types are only representative of pollution that may occur from shipping activities, or can also be used to understand the impacts of oil spills from drilling activities.*'

- Author response: We would like to highlight the sentence which immediately precedes the one raised above: L 237 '*Otremba established that, for the oils capable of forming a surface film, Romashkino crude oil has the largest imaginary refractive index and absorption coefficient values whereas Petrobaltic crude oil has the smallest values, with other oils bracketed by these parameters (Otremba, 2000).*'
- The aim of this study is an exploration of how oil can affect sea ice and according to Otremba (2000) these oils succinctly envelope the upper and lower optical properties of typical oils. It is feasible for both types of oil, and everything in between, to be in the marine realm owing to the various ways in which oil is released (e.g. subsurface release, oil spill etc.), the densities of the oil, and is justified in Otremba (2000). The focus here is on the presence of background mass ratios of oil and is not prescriptive of the method or release into the environment.

14. Reviewer comment: '*l275: There seems to be a repetition in the first and second sentences. In the second sentence, the author state the same elements as in the first sentence but with melting and multi-year sea ice in addition. Please fix or clarify.*'
- Author response: The text is correct as it is a repetition of the study, however it has now been reworded: '*Albedos of quasi-infinite first-year sea ice doped with different mass ratios of both Petrobaltic and Romashkino from 0–1000 ng g$_{-1}$ (0, 5, 10, 25, 100, 150, 200, 250, 300, 350, 400, 450, 500, and 1000 ng g$_{-1}$) are calculated as a function of wavelength. The study presented here is then repeated for quasi-infinite melting, first-year, and multi-year sea ice.*'

15. Reviewer comment: '*l279: In the sentence concerning the effect of oil droplet size, it would be useful to explicitly state the sea ice type considered.*'
- Author response: the text has been changed and now reads: '*The effect of oil droplet size (in the range 0.05–5.0 μm) are then considered for a single wavelength (400 nm) for the oil with largest absorption cross section (Romashkino) and for quasi-infinite sea ice with the largest penetration depth to light (melting sea ice) (Marks and King, 2014).*'

16. Reviewer comment: *'Figure 1: I suggest using a Y axis ranging from 0.3 to 0.9, and putting the legend outside the figure for more clarity, if this is allowed by editing rules.'*

- Author response: The authors believe it is more valuable to have the same x-axis for all figures to allow easy comparison. The raw data will be made available to allow accurate reading of any data.

17. Reviewer comment: *'Section 3.3 Could the authors specify why melting sea ice was chosen for the analysis of the effects of oil droplet size on albedo? Are the implications similar for other types of sea ice?.'*

- Author response: The melting sea ice was chosen as it has the largest penetration depth to light (Marks and King, 2014) and is justified in point 15 above.
- The implications are similar for the other types of sea ice, however melting sea ice is liable to have the least conservative effect and we therefore found it most useful to explore the significance of droplet size.
- We have added the following sentence to Section 3.3: *'Melting sea ice was selected as it is liable to have the least conservative effect owing to it having the largest light penetration depth of the three types of sea ice.'*
-

18. Reviewer comment: *'Section 3.4: A reference to ΔA/Δm used in Figure 5 is expected in the text.'*

- Author response: The text in Section 3.4 has been edited to now read: *'The upper row of Fig. 5 (A, C, and E) indicate albedo versus increasing mass ratios of Romashkino oil, whereas the lower row (B, D, and F) is a metric for the sensitivity of an ice to oil in the presence of black carbon, where sensitivity is the rate of change in albedo with increasing mass ratios of Romashkino oil (i.e. dA/dM).'*

19. Reviewer comment: *'l627: "Arctic multi-year and first-year sea ice are declining at 17.5% and 13.5% respectively..." is not clear. Please rephrase.'*

- Author response: The text has been edited to now read: *'Arctic multi-year and first-year sea ice extent are declining at 17.5% and 13.5% per decade respectively'*.

20. Reviewer comment: *'l641: "[...] this is the first instance that the climatic effect of oil pollution on sea ice has been considered." This sentence is misleading and implies the use of a climate model or conclusions on the large scale impact of oil pollution in the Arctic which is not the case here. Please rephrase.'*

- Author response: The text has been edited to now read: *'this is the first instance that the background effect of oil pollution on sea ice albedo has been considered.'*

21. Reviewer comment: *'l659: "[...] the findings of this study may only be valid during the ablation season when snow cover has melted or been removed by wind." In this case why consider different types of sea ice? I believe the value of this paper lies in the sensitivity study considering a variety of optical and physical parameters. I suggest to add that this may be the case in practise and that the authors restate the main purpose of the study..'*

- Author response: As suggested by the reviewer, we have added the following text to the end of Section 4.7 to restate the purpose: *'The paper presented here is a*

*sensitivity study considering a wide variety of optical and physical parameters for the oil pollution of sea ice. This is an exploratory study and will hopefully act as a foundation for more sophisticated studies coupled with field and lab-based experiments to follow.'*

---

## Author Response (AR1)

To the Editor (Yevgeny Aksenov)

Thank you for agreeing to serve as editor and offering the authors the opportunity to respond to the comments from the referees. We are pleased that both reviewers recognise the merit of this manuscript and are grateful that reviewer 2 recommended this work for publication after highlighting its value for the scientific community.

We have addressed all the comments from both referees and have been diligent in answering all points; please see our responses attached below. We have given particular attention to the concerns raised by reviewer 1 throughout the text and have added the additional figure 8 to alleviate the comments regarding scattering variability.

In our responses we have printed the reviewers' comments in black and our responses in blue. The underlined text immediately precedes quotations from the text that has been added or edited; the text that has been added or edited to the manuscript are italicised and between quotation marks.

Reviewer 1 response: 2-7 Reviewer 2 response: 8-16

Kind regards,

Ben Redmond Roche and Martin King

**Response to Anonymous Referee #1 on tc-2021-372**

- 1. Reviewer comment 'I find the title somewhat misleading, as the authors really have not quantified the climatic impact of oil pollution'.
- Author response: title changed to 'Quantifying the effects of background concentrations of crude oil pollution on sea ice albedo'.
- 2. Reviewer comment 'I don't believe that the optical descriptions given here for firstyear, multiyear, and melting sea ice are realistic or representative in the context of this study. It is well established that the scattering coefficients for sea ice display significant variability, including between ice types, within a single ice column, and for the same ice type at different times and locations.'.
- Author response: It is difficult to reply accurately to this comment as no comparison to data or references are provided by the reviewer.
- The authors are aware that scattering cross sections for sea ice vary within the ice column, with seasons and between geographic locations. We present a new figure in Section 4.7 of the paper using a range of scattering sections from literature (Grenfell and Maykut, 1977; Perovich, 1990, 1996; Timco and Frederking, 1996; Gerland et al., 1999; Simpson et al., 2002; Fisher et al., 2005; King et al., 2005; France et al., 2011; Marks and King, 2013) for typical thickness sea ices. This graph also shows that whilst there is variability in the albedo response of each type of sea ice, the conclusions put forward in this manuscript are valid:

**Fig. 8.** Wavelength dependent albedo of multi-year (A), first-year (B), and melting sea ice (C). Shown here is the variability in albedo for each type of sea ice based on scattering cross sections described in literature (Grenfell and Maykut, 1977; Perovich, 1990, 1996; Timco and Frederking, 1996; Gerland et al., 1999; Simpson et al., 2002; Fisher et al., 2005; King et al., 2005; France et al., 2011; Marks and King, 2013), ranging from 0.5–1 m2 kg-1 for multi-year sea ice; 0.1–2 m2 kg-1 for first-year sea ice; and 0.01–0.05 m2 kg-1 for melting sea ice. Shown in red are the data with no oil pollution present and shown in blue are the data with 1000 ng g-1 of *Romashkino* oil present. The thicker lines show the values of typical ice used in this study. The melting sea ice and multi-year sea ice are 2.5 m thick; the first-year sea ice is 0.8 m thick, and the background concentration of black carbon is set to 5.5 ng g-1.

- -
- Whilst we cannot know which scattering cross section or albedo values the reviewer is comparing to, we would like to make clear for the reader/reviewer that our study does the following:

- (1) lists monochromatic albedo which must be integrated to compare to broadband albedo. Reported broadband albedo values are thus lower.
- (2) Focuses on three different types of sea ice which are assigned a quasi-infinite thickness as sea ice thickness and type vary so much. The quasi-infinite thickness allows a fair 'like-for-like' comparison between optical properties; the three types of ice are simple examples to explore the variation in scattering cross sections as opposed to being rigid prototypes. The three different types of sea ice are based on the loci of their scattering cross sections.
- (3) The work is an exploration, possibly the first of how oil optical properties may affect sea ice and so the mass ratio of black carbon has been kept at a pristine level for quasi-infinite cases.
- (4) The variation of scattering cross sections within the ice column has not been explored as this is the first exploratory study to determine if oil is a concern for sea ice albedo. Further detailed models with a plethora of ice types and possibly dynamics are now all possible, but this initial study demonstrating the effect is important so that others can choose to explore these effects further. It should be stated that there has been significant interest in this unpublished manuscript from environmental and petrochemical organisations.
- We have added the following text to the manuscript:
- Edited caption in Section 2.1: '**Table 1**. TUV-snow model sea ice input parameters derived from literature (Grenfell and Maykut, 1977; Perovich, 1990, 1996; Timco and Frederking, 1996; Gerland et a.l, 1999) and previously used in other studies (e.g. King et al., 2005; France, 2008; France et al., 2011; Marks and King, 2013, 2014; Lamare et al., 2016).'
- Added to Section 4.7: 'Throughout this study the typical mid-range values for sea ice have been selected and are reported in Table 1. To assess any uncertainties that may arise from the variability of scattering cross sections reported in literature (Grenfell and Maykut, 1977; Perovich, 1990, 1996; Timco and Frederking, 1996; Gerland et al., 1999; Simpson et al., 2002; Fisher et al., 2005; King et al., 2005; France et al., 2011; Marks and King, 2013) Fig. 8 is additionally presented and indicates the response of realistic types of sea ices with both lower and higher scattering cross sections.'
- 3. Reviewer comment: 'I'm a bit confused by the FY, MY, melting classifications. If Arctic sea ice isn't melting, it is likely snow covered. Do the authors intend for this study to treat bare, non-melting FY, MY ice? And, if the snow is implicitly included, then the dynamics of snow-oil interactions need to be accounted for.'.
- Author response: We have done a general study for the effect of oil pollution on bare sea ice. Sea ice is frequently snow covered so this study, similar to the study of Light et al, 1998, may only be valid for areas of sea ice that experience snow melt or removal of snow by wind. Whilst this study focuses on the Arctic owing to the development of shipping routes and enormous hydrocarbon reserves in the region, these results are also relevant for non-polar sea ice and the Antarctic.
- The reviewer has raised two minor issues here: (a) First-year, multi-year, and melting sea ice classification; and (b) snow cover on sea ice.
- (a) Our first-year, multi-year, and melting sea ice classifications are a way of exploring the variation in scattering cross sections of sea ice as described in response

to point 2 above. We now explicitly state in the Uncertainties section that these are the mid-range/typical values selected from literature.

- (b) We have added the following text to the Section 4.7: 'Sea ice is frequently covered with snow and sea ice tends only to be free of snow where it is removed by katabatic winds or during the melting season (Weeks, 2010).'
- However, studies (Marks and King, 2013) have previously reported that an overlying snowpack of 2–5 cm is sufficient to mask light absorbing impurities in sea ice and so it was not necessary to consider snow cover in this work as it would mask our results. Unlike black carbon pollution in snow and sea ice, which is deposited from the atmosphere, the incorporation of background quantities of oil pollution into the sea ice matrix is postulated to come from the ocean. We have not considered the movement of oil from the sea ice matrix to snowpack in this work as it is an area of research and probably only relevant at much higher concentrations of oil.
- For clarity, the following has been added to Section 2.2: *'… snow cover… are not considered for the quasi-infinite sea ice in order to independently assess the effect that oil pollution has on the albedo of the different types of sea ice (Zdanowicz et al., 1998; Grenfell et al., 2002; Jiao et al., 2014).*'
- And: 'However, snow is not considered to allow for the effect of oil pollution on sea ice albedo to remain independent.'
- 4. Reviewer comment: 'L183: how do authors justify 201 layers?'
- Author response: Previous works (Marks, 2017) have demonstrated that a large number of layers are required at the interface, and 201 layers was a good compromise between computational time and precision.
- The following text has been added to Section 2.1: '*Marks (2017) found it important* to have a large number of layers at the interface so 201 layers offers a good compromise between computational time and precision.'
- 5. Reviewer comment: 'L200 (Eq1): What is the prime symbol for?'
- Author response: it was a comma and has been removed for clarity.
- 6. Reviewer comment: 'L212: why bother modelling atmosphere here? Seems extraneous'
- Author response: In this paper we report diffuse sky conditions to keep the work manageable. Our initial efforts included the effects of solar zenith angle and sky conditions (e.g. cloud, aerosol, ozone column etc.) but the work became too large for a single paper and the atmosphere radiative-transfer detracted from the sea ice radiative-transfer.
- 7. Reviewer comment: 'L299: "The oil is most absorbing at 400 nm, where ice is the least absorbing..." Perhaps this is true, but this is the cause, not a result.'
- Author response: we think there may be a misunderstanding here, we have taken the optical properties and done a Mie calculation on them this is a result and helps the reader understand the results more clearly.
- We are describing the absorption efficiency of an oil droplet (similar size to the wavelength of light) because that is different depending on the size (see Fig. 3) to the absorption cross section.

- For clarity, we have changed the text in Section 3.1 to now read: 'The Mie calculation demonstrates the oil droplets are most absorbing at a wavelength of 400 nm where ice is the least absorbing (Warren and Brandt, 2008).'
- 8. Reviewer comment: 'L313: "the effect of oil significantly decreases as wavelength increases over the region studied and the oil becomes less absorbing whilst the ice becomes more absorbing." Same as previous comment, this is a physical cause, not a result.'
- Author response: this statement resides from the fact that we have done a Mie calculation and is therefore a result.
- For clarity, we have changed the text in Section 3.1 to now read: 'the Mie calculations indicate the effect of oil significantly decreases as wavelength increases and the ice becomes more absorbing.'
- 9. Reviewer comment: 'L328: "The three types of sea ice have different unpolluted albedos: melting sea ice 0.72, first-year sea ice 0.87, and multi-year sea ice 0.94 at a wavelength of 400 nm respectively, owing to their different scattering cross sections (Perovich, 1996; Marks and King, 2014)." I think this overstates the differences between these three ice types if indeed it is intended that all are snow free.'
- Author response: the ice is snow free as now clearly stated in Section 2.2: *'… snow cover… are not considered for the quasi-infinite sea ice in order to independently assess the effect that oil pollution has on the albedo of the different types of sea ice (Zdanowicz et al., 1998; Grenfell et al., 2002; Jiao et al., 2014).*'
- And: 'However, snow is not considered to allow for the effect of oil pollution on sea ice albedo to remain independent.'
- This comment is a repeat of the issues responded to above in point 2 and our explanations and edits are again valid here.
- Please note these are the maximum monochrome albedo values of each type of sea ice taken at a wavelength of 400 nm and that these sea ice's have been made quasiinfinite in order to independently compare the effects of oil pollution on sea ice albedo.
- 10. Reviewer comment: 'L606: "As these data shows, this decline in perennial types of sea ice renders the Arctic much more vulnerable to increased oil pollution in the region..." I don't think this conclusion is supported by this study. For example, the high scattering prevalent in the surface layers of multiyear ice, and the larger thickness of this layer in thicker ice is not accounted for in this study. Also, there is no attempt to simulate how oil droplets respond to summer freshwater flushing that is a key factor that distinguishes FY ice from MY ice.'
- Author response: we have edited the text in Section 4.4: 'As these data show, this decline in perennial types of sea may render the Arctic more vulnerable to increased oil pollution in the region, particularly as it is opened to both shipping and oil extraction.'
- We have done the first study to show that background concentration levels of oil in sea ice are important – the model is a radiative transfer model only and allows for the effect of oil pollution to be determined and for the basis to be set for more complex work in the future utilising more sophisticated models.

- We have added a line to section 4.7: 'This manuscript is a sensitivity study that considers a wide variety of optical and physical parameters for the oil pollution of sea ice. Ultimately, the results indicate that background concentrations of oil pollution may have an important effect on sea ice albedo. It is anticipated that this study will act as a foundation for more complex studies to follow that, coupled with field and lab-based experiments, may explore the effects such as the movement of oil within the sea ice column in greater detail.'
- Martin (1979) explained that once oil reaches the ice surface it will be reintroduced to the ocean in a weathered or emulsified form and potentially be available to be reincorporated in newly forming ice we considered this and the fact that oil droplets will rise in the ice column as brine channels broaden in the spring/summer. It is not clear how it would be possible to model the effects of flushing, so we have not attempted to do so in this manuscript.
- We have alluded to this in the text and added the following line to Section 4.1: 'Martin (1979) also found that in summer oil at the ice surface leads to melt-pond formation owing to the absorption of solar energy; once on the surface, the oil will be reintroduced to the ocean in a weathered or emulsified form by melting through the ice or flowing of the sides.'
- 11. Reviewer comment: 'L621: "Therefore, it appears that the type of oil has the biggest effect on how responsive sea ice is to increasing mass ratios of oil as opposed to the type of sea ice and in contrast with the findings of a comparable study into the effects of mineral dust on sea ice albedo (Lamare et al., 2016)." I don't understand what this means. Is it saying that there is larger variability in oil inherent optical properties than in the optical properties of mineral dust? That may be so, but it is not a result or a conclusion.'
- Author response: we have edited the text in Section 4.5 for clarity: 'This differs from a similar study Lamare et al., 2016) which concluded that the optical properties of sea ice played a more important role on the response of sea ice albedo than the type of pollutant (e.g., windblow aerosols).'
- This is a discussion point, not a result or a conclusion, and is in the Discussion section (4).
- 12. Reviewer comment: 'L627: "First-year and particularly melting sea ice are more responsive to oil pollution than multi-year sea ice, so these trends indicate that sea ice albedo in the Arctic may become more vulnerable to background levels of oil 630 pollution as the ice becomes progressively thinner and younger." I find this conclusion unsubstantiated, because I don't think differences in the optical properties of FY / MY ice types is treated in a realistic way here. This simplification may be well justified for the purposes of a sensitivity study such as carried out here, but I think it's a stretch to draw conclusions such as stated here from this type of sensitivity exercise.'
- Author response: we have demonstrated in our comments to point 2 above that the optical properties of first-year and multi-year sea ice used in this manuscript indeed are valid and based on measurements taken from literature (Grenfell and Maykut, 1977; Perovich, 1990, 1996; Timco and Frederking, 1996; Gerland et al., 1999; Simpson et al., 2002; Fisher et al., 2005; King et al., 2005; France et al., 2011; Marks and King, 2013).

- We have edited the text in Section 4.6: 'First-year and particularly melting sea ice may be more responsive to oil pollution than multi-year sea ice, so these trends indicate that sea ice albedo in the Arctic may become more vulnerable to background levels of oil pollution as the relative amount of these types of ice dominate in the Arctic.'

**Response to Anonymous Referee #2 on tc-2021-372**

- 1. The authors would like to thank the reviewer for their positive comments and the recommendation that this manuscript should be published.
- 2. Reviewer comment: 'There is a misalignment between the title of the article and certain statements (see detailed comments below) and the actual purpose of the proposed work. Indeed, the climatic impact of crude oil pollution is not addressed in this sensitivity study. I would recommend that the authors rephrase the title of the manuscript and correct the statement line 641.'
- Author response: The title has been changed to: 'Quantifying the effects of background concentrations of crude oil pollution on sea ice albedo.'
- The reviewer raises a valid point, the full climatic potential has not been realised by our paper as we only explore albedo. We have therefore removed the word climatic from line 641.
- 3. Reviewer comment: 'The impacts of crude oil droplets dispersed in sea ice are investigated in the visible wavelengths (400-700 nm). Could the authors please expand on the reasons for this wavelength range? Is it a limitation of the model or a deliberate choice? Most observation tools (ground-based instruments, drone or aircraft mounted sensors, Earth Observation satellite) cover a larger spectrum, generally from the visible to the shortwave infra-red. Furthermore, climate models (e.g. CMIP models) consider the longwave radiative balance to monitor the Earth's energy balance. Although longwave radiation is most likely out of scope for this study, it is surprising that such a small range of wavelengths is being considered here. Extending the range to 2500 nm would allow direct comparisons with observations.'
- Author response: The selected wavelengths are a deliberate choice by the authors as we are only measuring over the wavelength window where oil has an effect on the albedo of sea ice.
- To clarify this for the reviewer we have attached a figure below comparing the absorption coefficients of oil taken from Warren and Brandt (2008) to background levels of Romashkino oil between 10–1000 ng g-1:

---

## Author Response (AR3)

To the Editor (Yevgeny Aksenov),

Thank you for agreeing to serve as editor and offering your positive comments for these minor edits. Thank you for also agreeing to extend our deadline as Martin and I have been abroad. Please see below our responses to the remaining comments from Editor 1.

In our responses, we have printed the reviewers' comments in black and our responses in blue. The underlined text immediately precedes quotations from the text that has been added or edited; the text that has been added or edited to the manuscript are italicised and between quotation marks.

\*I have noticed that the line numbers I have mentioned below are incorrect in the Track Changes file but correct in the submitted final file. Sorry about this, it appears to be a common Track Changes error on Word, however, all of the edits are highlighted in red in the text and in the margin.

Kind regards,

Ben Redmond Roche and Martin King

**Response to Anonymous Referee #1 on tc-2021-372**

- 1. Reviewer comment 'If it is the authors' intent that first-, multiyear, and melting ice types be distinct ice types, and if no snow cover is being considered, then it needs to be explicitly stated that the first- and multiyear ice cases represent bare ice (without snow cover) that is not melting'.
- Author response: We have now made clear that it is bare sea ice in the Abstract, Methodology, and Uncertainties sections.
- line 8 changed in the Abstract to now read: 'In this study, the albedo response of three different types of bare sea ice (melting, first-year, and multi-year sea ice) are calculated at increasing mass ratios (0–1000 ng g-1) of crude oil by using a coupled atmosphere-sea ice radiative-transfer model (TUV-snow) over the optical wavelengths 400–700 nm.'
- Line 172 changed in Section 2.1 to now read: 'In this study, the albedo response of three different types of bare sea ice (multi-year sea ice, first-year sea ice and, melting sea ice) to increasing amounts of two different crude oils are calculated using the TUV-snow model.'
- Line 685 changed in Section 4.7 to now read: '*Therefore, similar to the work by Light et al (1998), the findings of this study may only be valid during the ablation season when snow cover has melted or been removed by wind, as the different types of sea ice considered here are bare.*'
- 2. Reviewer comment 'Statements such as are included in the abstract (line 22 24) are thus difficult to justify: "All three types of sea ice are affected, however firstyear sea ice and particularly melting sea ice are very sensitive to oil pollution; thus, the Arctic may become more vulnerable to oil pollution as the ice becomes progressively thinner and younger in response to a changing climate."
- Additionally, the conclusion (line 764) that "*This may become increasingly important as multi-year sea ice continues to decrease, first-year sea ice becomes more dominant and there is earlier melting of sea ice in the year*." is also not justified, since the first- and multiyear ice types considered in this study are anomalous, not the more common melting variety.

- Author response: We have made the following changes in the Abstract and Conclusion sections.
- Line 22-24 in the Abstract removed, line 20 changed to now read: '*Therefore, the work presented here demonstrates that low background concentrations of small submicron to micron-sized oil droplets have a significant effect on the albedo of bare sea ice. All three types of bare sea ice are sensitive to oil pollution, however first-year sea ice and particularly melting sea ice are very sensitive to oil pollution.*'
- Line 732 in the Conclusion, referred to by the reviewer as line 764, has now been removed.
- **3.** Reviewer comment 'One could argue that since snow covered sea ice is not considered in this study, the only significant ice type to draw climate-relevant conclusions from would be the melting ice. I understand that the properties of these three ice types were adopted from earlier work (Marks and King, 2014), and it is acceptable to run these cases through this sensitivity analysis, but it is an overstep to draw conclusions about the implications of these studies on the climate.'.
- Author response: Our Abstract, Discussions, and Conclusions sections no longer discuss or mention climate. The world climate only appears in the Abstract and Introduction sections (line 1 and 42) to explain how albedo is an important component of Earths climate.

**Minor/technical comments**

- 4. Reviewer comment: 'Title: add 'Arctic' sea ice albedo'.
- Author response: These results are not specific only to the Arctic, so we decline to add the word Arctic to our title as our findings are applicable to all bare sea ices on Earth.
- 5. Reviewer comment: '30-31: "Perennial sea ice cover decline is between 12.2% and 13.5% for first-year sea ice..." Confusing. Usually "perennial" sea ice is considered ice that survives at least one summer melt season. Maybe "Interannual sea ice cover decline..." was intended'
- Author response: Thank you for highlighting this error, it has now been corrected.
- Line 30 in section 1 changed to now read: '*First-year sea ice cover is declining at between 12.2% and 13.5%, and multi-year sea ice cover is declining at between 15.6% and 17.5% per decade, respectively (Comiso, 2012; Tschudi et al., 2019).*'
- 6. Reviewer comment: '41 42 "The high latitude, radiative balance, is..." delete commas; also, this applies only in summer.'
- Author response: Thank you for highlighting this error, it has now been corrected.
- Line 42 in section 1 changed to now read: *'The high-latitude radiative balance is primarily controlled by shortwave solar radiation during the summer which significantly affects both sea ice and snow cover in the region (e.g., Perovich et al., 1998; Flanner et al., 2007).'*
- 7. Reviewer comment: '54 56: "The albedo of sea ice is wavelength dependent with maximum albedo values occurring at 390 nm in pure ice, where absorption is at a minimum (Warren at al., 2006)." I agree this is the theoretical maximum albedo, but I've never seen an observation that shows this is true for sea ice.'
- Author response: Sentence unchanged as the reviewer appears to agree with us.
- 8. Reviewer comment: '*Table 1: If the properties of the ice cover are resolved by multiple vertical layers in the model, why are the properties of the ice uniform? It is well*

established that 3 layers are needed to simulate radiative transfer in sea ice (see e.g., Light et al., 2008)'

- Author response: The three types of ice described in this study are based on previous work (e.g., Marks and King, 2013; 2014; Lamare et al., 2016), which are themselves based on the seminal work by Grenfell and Maykut (1977). We agree that other studies such as Light et al (2008) have established the three layers and have now added reference to this in the text. It is also unknown how oil moves between different types of ice fabric, so we have therefore kept the optical properties of the ice homogeneous.
- Text added to Section 2.1: '*The types of ice have been kept simple and in keeping with previous work (e.g., Marks and King, 2013; 2014; Lamare et al., 2016), further studies with a better understanding of how oil moves between different ice fabrics may wish to consider the three-layer model proposed by Light et al (2008).*'
- 9. Reviewer comment: '*Eqn(2), fine, but sigma traditionally used for scattering coeff, not absorption coeff. But OK since it's defined.*
- Author response: Different fields use different nomenclature; we are using the same symbols as the original radiative transfer model (Lee Taylor and Madronich, 2002).
- 10. Reviewer comment: "… ice sheet continues to grow underneath the sea ice,…" what does this mean? Maybe it should be " …and the ice continues to grow at its interface with the ocean"?'
- Author response: Whilst the original text makes sense, it has now been clarified for the reader.
- Line 516 of Section 4.1: 'Oil is known to entrain itself into sea ice in several ways: (1) oil beneath the sea ice is frozen into the sea ice and the ice continues to grow at its interface with the ocean, with the oil being fully encapsulated in the ice matrix within 18–72 hours'
- 11. Reviewer comment: '551: which require more work—what does this mean'
- Author response: The text has now been edited.
- Line 529 of Section 4.1: '*These studies have focused on macroscopic quantities of oil* whereas the work presented here focuses on microscopic-sized background concentrations of oil which are less well characterised.'
- 12. Reviewer comment: 663: *"relatively light absorbing" is potentially confusing—does it mean relatively minimally absorbing, or does it mean relatively energy absorbing?*
- Author response: The text has now been edited.
- Line 717 of Section 5: '*The albedo response is dependent on the type of oil, with the strongly absorbing Romashkino oil (relative to Petrobaltic oil) having the largest effect owing to its large mass absorption coefficient.*'